# A Geometric Analysis of Small-sized Language Model Hallucinations

Emanuele Ricco [1]  Elia Onofri [1]  Lorenzo Cima [2 3]  Stefano Cresci [2]  Roberto Di Pietro [1]

## Abstract

Hallucinations—plausible but factually incorrect responses—pose a major challenge to the reliability of Large Language Models (LLMs), especially in multi-step or agentic settings. Existing work largely frames hallucinations as a consequence of missing knowledge; we show instead that, even when the relevant factual knowledge is present, models still produce hallucinated answers, pointing to retrieval instability rather than knowledge gaps. Building on this observation, we introduce APORIA (*Aggregate Prompt-wise Observation Retrieving Instability via Asymmetry*—the Socratic state of "puzzlement-in-contradiction" that hallucinations embody), a geometric framework that studies repeated responses to the same prompt in sentence-embedding space. Our central hypothesis is that genuine responses cluster more tightly than hallucinated ones; we empirically validate this and show that, after Fisher projection, the two response classes become consistently separable. We leverage this asymmetry in geometry via APORIA-LP, an efficient label-propagation method that classifies large collections of responses from as few as 30–50 annotations, achieving F1 scores above $90\%$ across ten small-sized LLMs. To support further research, we release SOCRATES-300K, a fully labelled dataset of 300,000 responses, together with the code for both dataset generation and result reproduction. Our key finding—framing hallucinations from a geometric perspective in the embedding space—complements traditional knowledge-centric and single-response evaluation paradigms, paving the way for further research.

[1]Computer, Electrical and Mathematical Sciences and Engineering (CEMSE) division, King Abdullah University of Science and Technology (KAUST), Thuwal, Saudi Arabia [2]Istituto di Informatica e Telematica (IIT), National Research Council of Italy (CNR), Pisa, Italy [3]Department of Information Engineering, University of Pisa, Pisa, Italy. Correspondence to: Elia Onofri <elia.onofri@kaust.edu.sa>.

*Proceedings of the 43$^{rd}$ International Conference on Machine Learning*, Seoul, South Korea. PMLR 306, 2026. Copyright 2026 by the author(s).

## 1. Introduction

Language models are now extensively deployed across a wide range of applications, including information retrieval, content generation, decision support, and interactive systems. Alongside large, resource-intensive models accessible primarily to well-resourced organisations, *small-sized language models* have gained increasing adoption due to their efficiency, deployability, and transparency (Wu et al., 2025). Advances in scaling laws and training strategies have further narrowed the performance gap between small models and earlier generations of much larger systems (Chung et al., 2024), making them attractive in practical pipelines and agentic architectures (Belcak et al., 2025; Wang et al., 2025). In such settings, reliability becomes critical: incorrect or fabricated responses can propagate through multi-step reasoning chains or coordinated agents, amplifying their impact (Hao et al., 2024).

A central obstacle to reliability is the generation of responses that appear fluent and plausible while being factually incorrect, commonly referred to as *hallucinations* (Huang et al., 2023; Ji et al., 2023; Li et al., 2023; Chen et al., 2023; Sahoo et al., 2024). Much of the existing literature frames hallucinations as a consequence of missing or inaccessible knowledge, often attributing incorrect generations to limitations in training data coverage, probabilistic decoding, or noise in internal representations (Lin et al., 2022; Xu et al., 2024; Kalai et al., 2025). As a result, substantial effort has focused on hallucination detection and evaluation, typically treating each prompt–response pair individually while asking if the model "knows" the correct answer (Farquhar et al., 2024; Niu et al., 2025; Su et al., 2024; Tonmoy et al., 2024).

However, this single-response perspective obscures an important empirical observation: for many prompts, the same model can generate both correct and incorrect answers across repeated samplings. In such cases, the relevant factual knowledge is arguably present, yet generation might diverge toward incorrect outcomes. This suggests an alternative interpretation where hallucinations are not primarily driven by absence of knowledge, but by failures in how available information is retrieved and instantiated at generation. From this perspective, hallucinations emerge *not* as isolated errors, but as part of a broader *structural organisation* of responses produced under the same prompt.

In this work, we adopt this alternative viewpoint and we study hallucinations through a geometric analysis of repeated model responses. Rather than analysing internal model states or the generation process itself, we focus on the *emerging relations within the space spanned by the embeddings of responses*: we embed multiple generations obtained under the same prompt and examine the distributional structure formed by genuine and hallucinated answers. We restrict attention to prompts for which models produce multiple factually correct responses, ensuring that the analysis isolates hallucinations arising under conditions where the required knowledge is demonstrably available. Under this assumption, divergence across generations can be interpreted as a retrieval instability (or instantiation-level phenomenon) rather than a persistent misconception.

Our analysis reveals that genuine and hallucinated responses exhibit systematically different geometric properties in embedding space, with hallucinated responses displaying reduced semantic cohesion relative to genuine ones. These structural differences are stable across models, prompts, and class balances, and cannot be explained by sampling artefacts alone. Moreover, we show that this geometry can be exploited to propagate labels from a small set of judged responses to large collections of prompt-consistent generations, enabling accurate and label-efficient classification.

Taken together, our results suggest that hallucinations in small-sized LLMs can be effectively detected as a geometric phenomenon arising from retrieval instability, rather than solely as a lack of knowledge. This shift in perspective complements existing detection-focused approaches and highlights the value of analysing hallucinations at the level of response distributions rather than individual generations.

**Contributions** Our key contributions are as follows:

**(i)** We formulate and validate the hypothesis that hallucinated responses arising under conditions of available factual knowledge exhibit, in the embedding space, weaker semantic cohesion than non-hallucinated ones, displaying distinctive and stable geometric signatures.
**(ii)** We introduce APORIA (*Aggregate Prompt-wise Observation Retrieving Instability via Asymmetry*; a framework echoing the Socratic *aporía*, the puzzlement-in-contradiction that captures the very phenomenon of hallucination). Leveraging the above geometric interpretation, APORIA attains strong distributional separability between genuine and hallucinated responses.
**(iii)** We exploit this separability to introduce APORIA-LP, a scalable, geometry-aware label-propagation framework that transfers hallucination tagging from a small set of judged responses to large collections of prompt-consistent generations, achieving F1 scores above 90%.
**(iv)** We release a fully labelled dataset of repeated responses (150 generations for 200 prompts across 10 LLMs) under

the name SOCRATES-300K to support structural analyses of hallucinations[1]. We also release the full code base for both dataset generation[2] and result reproduction[3].

## 2. Related Work

Small-sized LLMs (7–30B) offer practical advantages in efficiency, deployability, and accessibility, enabling fast inference and domain-specific customisation. However, these benefits raise important concerns about reliability and trustworthiness—particularly regarding hallucinations—motivating the need for systematic analysis and detection strategies (Bang et al., 2025).

**Trustworthiness in Small Language Models** Trustworthiness in language models is often discussed in terms of factual accuracy, calibration, and reliability under uncertainty. TrustLLM is a relevant framework in this field (Huang et al., 2024), establishing eight trustworthiness dimensions, including truthfulness, safety, and robustness, and evaluating them across 16 models and 30+ datasets. Trustworthiness is a broad term encompassing safety (Qi et al., 2025), alignment (Ouyang et al., 2022), and ethical considerations (Bender et al., 2021), with recent work focusing on empirical evaluation frameworks (Liang et al., 2024).

Factuality evaluation benchmarks provide a standardised assessment of truthfulness across diverse scenarios: FactBench (Fatahi Bayat et al., 2025) introduces a benchmark with 1K prompts spanning 150 topics, revealing that factuality does not necessarily improve with scale; FACTS Leaderboard (Jacovi et al., 2025) evaluates language models' ability to generate factually accurate responses in information-seeking scenarios; FactScore (Min et al., 2023) decomposes long-form generations into atomic facts supported by reliable sources, while Ling et al. (2024) investigates trustworthiness of LLMs associated with in-context learning. Beyond individual response accuracy, recent work has identified systematic homogeneity in model outputs, with $> 70\%$ similarity observed on different models (Jiang et al., 2025).

**Hallucination Detection** Recent advances in hallucination detection exploit internal model representations rather than relying solely on output-level analysis. Sriramanan et al. (2024) investigates LLM internal states while Han et al. (2024) explore neuron-level mechanisms identifying specific activation patterns correlated with hallucination behaviour. Building on these approaches, Bar-Shalom et al. (2025) design a Vision Transformer-inspired model while Niu et al. (2025) propose a methodology based on critical token selection from a response. Conversely, Su et al. (2024) introduce an unsupervised framework leveraging internal

---

[1]Available on Zenodo — DOI:10.5281/zenodo.20256462
[2]Available on GitHub — emarich/Socrates-300K
[3]Available on GitHub — eOnofri04/APORIA

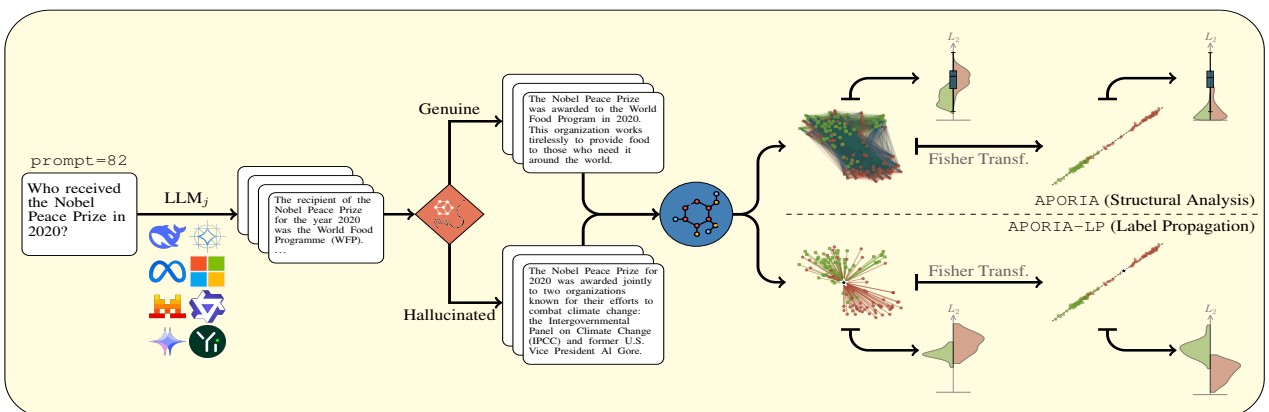

*Figure 1.* Pipeline of the proposed method. (Left) 200 different prompts are fed into 10 LLMs, generating 150 responses each. (Center) Responses are tagged as Genuine (G, green) or Hallucinated (H, red) under Claude Sonnet 4.5, and embedded through SBERT (blue circle) in $\mathbb{R}^{384}$ (here reduced with T-SNE for plots). Two analyses follow. *(Structural, top right)* In APORIA, mutual distances intra-G, intra-H, and inter-(GH) (blue) are evaluated in the embedding space, showing significant differences in the distributions (cf. Figures 2 and 3), yet being non directly separable; data points are embedded into the linear space maximising Fisher criterion, achieving optimal separability in terms of spatial and distances distribution (cf. Table 1). *(Label Propagation, bottom right)* In APORIA-LP, the structural analysis is exploited to classify previously unseen responses generated under the same setting: here, separability in the embedding space is potentially scarce, requiring the usage of Fisher space for reliable performances (cf. Table 2): points are classified by the distributions of the point-to-clusters distances.

states of LLMs for real-time hallucination detection. Finally, Chen et al. (2024) present INSIDE, an unsupervised method that exploits a small set of stochastic responses to a fixed prompt and combines them with the model's internal hidden states to score the greedy response via an EigenScore derived from the spectrum of the sample covariance.

Along with internal representations, semantic-level approaches analyse the meaning and consistency of responses without typically relying on white access to the model. Amongst these, two methods analysing plain responses are particularly relevant: Farquhar et al. (2024) propose SemanticEntropy, a framework that samples several answers to each prompt and clusters them into answers that have similar meanings. Conversely, Manakul et al. (2023) investigate the possible contradiction between hallucinated responses introducing SelfCheckGPT, a sampling-based approach that cross-checks the answers of multiple LLMs. Both works (alongside INSIDE) release open-source code and are later used in Section 4.2 as external baseline comparisons. Unfortunately, no suitable datasets are released, which motivates the introduction of SOCRATES-300K; the sole exception is INSIDE, whose dataset is not released yet remains reproducible (cf. Appendix D).

Overall, most existing work treats hallucinated responses as a consequence of missing or incorrect knowledge. In contrast, our work focuses on questions for which the model demonstrably possesses the relevant knowledge and investigates the structural patterns in representation space that nonetheless lead to hallucinated generations.

## 3. Methodology

We now describe the framework used to analyse response variability and hallucination behaviour. Our approach is organised around three complementary components: (i) the SOCRATES-300K dataset (see also Appendix A.1); (ii) the APORIA framework—a structural analysis of the embedded responses—which characterises geometric and distributional differences between genuine and hallucinated outputs; and (iii) the APORIA-LP label propagator, which extends annotations from a small set of responses to much larger collections without additional supervision.

### 3.1. SOCRATES-300K

Our analysis focuses on factual hallucinations, where responses contain false or misleading information presented as facts (Sun et al., 2024). To this end, we adopt time-sensitive questions to assess the trustworthiness of LLMs on evolving factual queries (Zhu et al., 2025). This choice is motivated by empirical evidence that smaller models exhibit substantially weaker performance on time-sensitive questions, making them more susceptible to hallucinations about temporal events (Pezik et al., 2025). Furthermore, these models show increased reliability with repeated sampling (Brown et al., 2024), but existing analyses do not provide a quantitative metric for hallucination rates.

To build SOCRATES-300K, we employ 10 small-sized (7B to 32B) LLMs with publicly available weights[4] selected to

---

[4]Namely: deepseek-llm-7b-base, Mistral-7B-v0.1, Llama-3.1-8B, Yi-1.5-9B, Gemma-2-9b, SOLAR-10.7B-v1.0, Phi-4,

represent a range of model families, architectural variations, and parameter scales, while maintaining comparability in size. Crucially, all models are base variants, *i.e.*, pre-trained models without instruction or alignment fine-tuning, ensuring that observed behaviours reflect fundamental generation properties rather than task-specific conditioning.

We crafted our prompts to target precise information about 100 repetitive events occurring in 2020 and 2022, allowing us to analyse how factual knowledge stability evolves (Zhu et al., 2025) within a controlled setting, where both years fall within the training boundaries of all evaluated models (Pezik et al., 2025). For each model-prompt pair, we generate 150 independent responses under identical sampling conditions, yielding a total of 300,000 responses,

$$\textit{i.e.} \ \overbrace{100}^{\text{prompts}} \times \overbrace{2}^{\text{years}} \times \overbrace{150}^{\text{responses}} \times \overbrace{10}^{\text{models}} = 300{,}000\,,$$

of which 231,473 (77.16%) remain after filtering for class imbalance and annotation uncertainty (cf. Table 7).

To study the impact of representation on geometric structure, we analyse each response under two embedding regimes: (i) complete responses, embedded directly using the sentence encoder; and (ii) stemmed responses, obtained by applying morphological stemming to reduce lexical variation while preserving semantic content. In what follows, we focus on complete responses, while results related to stemmed responses can be found in Appendix C.1.

**LLM-as-a-Judge Evaluation** We adopt an LLM-as-a-judge strategy to categorise model responses as *Genuine* (G, factually correct), *Hallucinated* (H), or *Unknown* (U), where the latter corresponds to responses that explicitly acknowledge uncertainty or lack of knowledge. Large-sized LLM judging has recently emerged as a scalable alternative to human annotation for assessing factual correctness, reasoning quality, and independent evaluation of model outputs (Zheng et al., 2023; Pezik et al., 2025), demonstrating strong correlation with human judgements at a fraction of the cost (Liu et al., 2023). We employ this paradigm using a fixed, high-capacity, proprietary model (Claude 4.5 Sonnet), treated here as an external oracle. Responses labelled as *unknown* (0.73%) are excluded from the present analysis, allowing us to focus on the geometric and statistical properties of factually grounded versus hallucinated generations. The annotation enables the study of distributional structure, separability, and failure behaviour, also supporting the propagation of annotations from a limited number of judged responses to larger sets of generated responses.

To assess the reliability of the automatically assigned labels, we conducted a manual validation. A total of 1,000 instances ($\approx 0.4\%$ of the full dataset) were independently annotated by two computer science PhD students. Of these,

Qwen2.5-14B, Gemma-2-27b, and Qwen2.5-32B.

800 instances were annotated separately (i.e., 400 per annotator), while 200 instances were annotated by both. On the shared subset, the annotators agreed on 94% of the cases (Cohen's $\kappa = 0.86$), indicating strong agreement. Disagreements were adjudicated by a senior researcher acting as an independent super-annotator. Comparing the finalised human annotations against Claude's labels on the full set of 1,000 instances shows that Claude's annotations match the human consensus in 87.90% of the cases, supporting the accuracy and suitability of Claude-generated labels. We refer to Appendix A.2 for further details.

### 3.2. Notation and Setting

Let $q$ denote a fixed prompt and $\mathcal{R} = \{r_1, \ldots, r_N\}$ a set of $N$ responses generated independently by an LLM under identical decoding conditions. Each response is embedded using a fixed sentence encoder $\phi$, yielding vectors $\mathbf{x}_i \in \mathbb{R}^d$, where $d$ is the embedding dimension; unless stated otherwise, $\phi$ is instantiated by the sentence transformer SBERT all-MiniLM-L6-v2 ($d = 384$).

A subset $\mathcal{R}_t \subseteq \mathcal{R}$ of size $N_t \leq N$ is assigned semantic correctness labels $y_i \in \{G, H\}$, corresponding to genuine and hallucinated responses, respectively. We denote by $\mathcal{X}_{\text{G}}$ and $\mathcal{X}_{\text{H}}$ the sets of embedded responses corresponding to genuine and hallucinated outputs from $\mathcal{R}_t$.

### 3.3. Structural Analysis of Response Geometry

Our analysis is motivated by the empirical hypothesis that, for a given prompt, genuine responses exhibit greater semantic consistency than hallucinated responses. Informally, correct answers tend to concentrate around a stable semantic core, while hallucinations manifest greater variability, reflecting distinct fabricated or erroneous explanations. This suggests that genuine and hallucinated responses should exhibit different geometric signatures in embedding space.

To test this assumption, we introduce APORIA, a framework to analyse the pairwise distances within the labelled responses. More specifically, for each pair of embedded responses $\mathbf{x}_i, \mathbf{x}_j$, we collect the Euclidean distance $\|\mathbf{x}_i - \mathbf{x}_j\|_2$, and we derive three empirical distributions: (i) intra-genuine distances $\mathcal{D}_{\text{GG}}$, (ii) intra-hallucinated distances $\mathcal{D}_{\text{HH}}$, and (iii) inter-class distances $\mathcal{D}_{\text{GH}}$. Rather than summarising these distributions through low-order moments, we compare them using the one-dimensional Wasserstein distance. Specifically, we quantify separation by measuring the Wasserstein distance between $\mathcal{D}_{\text{GG}}$ and $\mathcal{D}_{\text{HH}}$ and comparing it to the same distance evaluated on the null hypothesis $H_0$ obtained by randomly permuting class labels. This allows us to assess both the statistical significance of observed differences and their deviation from randomness.

To complement the distributional analysis, we assess linear

separability between genuine and hallucinated responses using the Fisher Discriminant Analysis (FDA); a comparison between FDA and other non-linear projection methods is presented in Appendix F. Given the labelled embeddings $\mathcal{X}_{\text{G}} \cup \mathcal{X}_{\text{H}}$, we compute the class means $\boldsymbol{\mu}_G$ and $\boldsymbol{\mu}_H$, and the within-class scatter matrix

$$\mathbf{S}_W = \sum_{\mathbf{x} \in \mathcal{X}_{\text{G}}} (\mathbf{x}-\boldsymbol{\mu}_G)(\mathbf{x}-\boldsymbol{\mu}_G)^\top + \sum_{\mathbf{x} \in \mathcal{X}_{\text{H}}} (\mathbf{x}-\boldsymbol{\mu}_H)(\mathbf{x}-\boldsymbol{\mu}_H)^\top .$$

To ensure numerical stability in high-dimensional settings, we adopt the regularised formulation $\mathbf{S}_W^{(\lambda)} = \mathbf{S}_W + \lambda\mathbf{I}_d$, where $\lambda > 0$ controls the strength of regularisation. The Fisher discriminant direction is then given, up to scale, by $\mathbf{v} \propto \left(\mathbf{S}_W^{(\lambda)}\right)^{-1}(\boldsymbol{\mu}_G - \boldsymbol{\mu}_H)$. Projection onto $\mathbf{v}$ yields a one-dimensional representation in which genuine and hallucinated responses can be directly compared, and mutual distances are moved from $\mathcal{D}_{\text{GG}}$ ($\mathcal{D}_{\text{HH}}$, $\mathcal{D}_{\text{GH}}$) to $\Delta_{\text{GG}}$ ($\Delta_{\text{HH}}$, $\Delta_{\text{GH}}$). This step serves as a diagnostic tool, revealing whether the two response types occupy distinct regions of embedding space under linear projections.

### 3.4. Supervised Label Propagation

Building on the structural properties revealed by the previous analysis, we define, under the name of APORIA-LP, a supervised procedure to propagate labels from a small annotated subset $\mathcal{R}_t$ to the remaining responses in $\mathcal{R} \setminus \mathcal{R}_t$.

Using the Fisher direction $\mathbf{v}$ estimated from $\mathcal{R}_t$, each embedded response $\mathbf{x} \in \mathbb{R}^d$ is mapped to a scalar coordinate $z = \mathbf{v}^\top \mathbf{x}$. This projection induces a one-dimensional representation in which distances reflect discriminative variation between genuine and hallucinated responses. Then, for a given unlabelled response $r$ with projection $z$, we compute the distances to the projected points of each labelled class, yielding the point-to-set distance collections $\Delta_{\text{G}}(z) = \{|z - z_i| : z_i \in \mathcal{Z}_{\text{G}}\}$ and $\Delta_{\text{H}}(z) = \{|z - z_i| : z_i \in \mathcal{Z}_{\text{H}}\}$, where $\mathcal{Z}_{\text{G}}$ and $\mathcal{Z}_{\text{H}}$ denote the projection of $\mathcal{X}_{\text{G}}$ and $\mathcal{X}_{\text{H}}$.

These collections define empirical distance distributions, which, consistently with the structural analysis above, are compared to the corresponding intra-class distance distributions using the Wasserstein distance. Intuitively, a response is assigned to the class for which its induced distance distribution is more consistent with the internal structure of that class. In other words, we assign $r$ to the class minimising the Wasserstein discrepancy between $\Delta_\circ(z)$ and the reference intra-class distribution $\Delta_{\circ\circ}$, or, in formulas, the response is classified as Hallucinated if $W(\Delta_{\text{GG}}, \Delta_{\text{G}}) > W(\Delta_{\text{HH}}, \Delta_{\text{H}})$.

While this procedure resembles classical linear discriminant classification under Euclidean distance, its formulation in terms of distributional consistency aligns naturally with the structural analysis introduced for APORIA. This perspective allows classification decisions to be interpreted geometri-cally, in terms of conformity with the observed internal organisation of response clusters, rather than solely in terms of pointwise distance to a decision boundary.

## 4. Results

We now present the empirical results obtained by applying APORIA and APORIA-LP to SOCRATES-300K.

### 4.1. APORIA Structural Analysis

Figure 2 reports the empirical distributions of pairwise distances within and across response classes, computed in the sentence embedding space. For each model–prompt pair, we consider three distributions: intra-genuine distances $\mathcal{D}_{\text{GG}}$, intra-hallucinated distances $\mathcal{D}_{\text{HH}}$, and inter-class distances $\mathcal{D}_{\text{GH}}$. The comparison between $\mathcal{D}_{\text{GG}}$ and $\mathcal{D}_{\text{HH}}$ is statistically assessed using a Wilcoxon rank-sum test, whose outcome is reported above each panel.

Across all models, we observe a consistent structural asymmetry between genuine and hallucinated responses. In particular, intra-genuine distances tend to be smaller than inter-class distances, indicating that factually correct answers exhibit higher semantic cohesion. This difference is statistically significant in the large majority of cases, often at the $p < 10^{-3}$ level. However, despite this consistency, the distributions remain substantially overlapping: the inter-class distances (blue boxplots) are often entangled with both $\mathcal{D}_{\text{GG}}$ and $\mathcal{D}_{\text{HH}}$, especially in the presence of class imbalance. This highlights an important limitation of directly exploiting pairwise distances in the original embedding space: although a structural signal exists, it is not readily separable without additional information.

**Comparison Against Null Hypothesis** The robustness of this structural signal is further illustrated by the comparison against a null hypothesis $H_0$. Figure 3 shows a representative example for qwen2.5-32B, where prompts are ordered according to the observed Wasserstein distance between $\mathcal{D}_{\text{GG}}$ and $\mathcal{D}_{\text{HH}}$. For each prompt, the observed distance is compared with the distribution obtained under $H_0$, constructed by randomly permuting class labels. For the vast majority of prompts, the observed Wasserstein distance exceeds the null expectation by a substantial margin, often by multiple factors. This confirms that the separation between genuine and hallucinated responses is not an artefact of sample size or marginal distributions, but reflects a genuine geometric difference in embedding space. A complete quantitative summary across models and prompts is provided in Appendix C.

**Fisher Discriminant Analysis** While statistically robust, this separation remains difficult to exploit directly. To make the structural difference operational, we project the responses onto the one-dimensional subspace identified by the

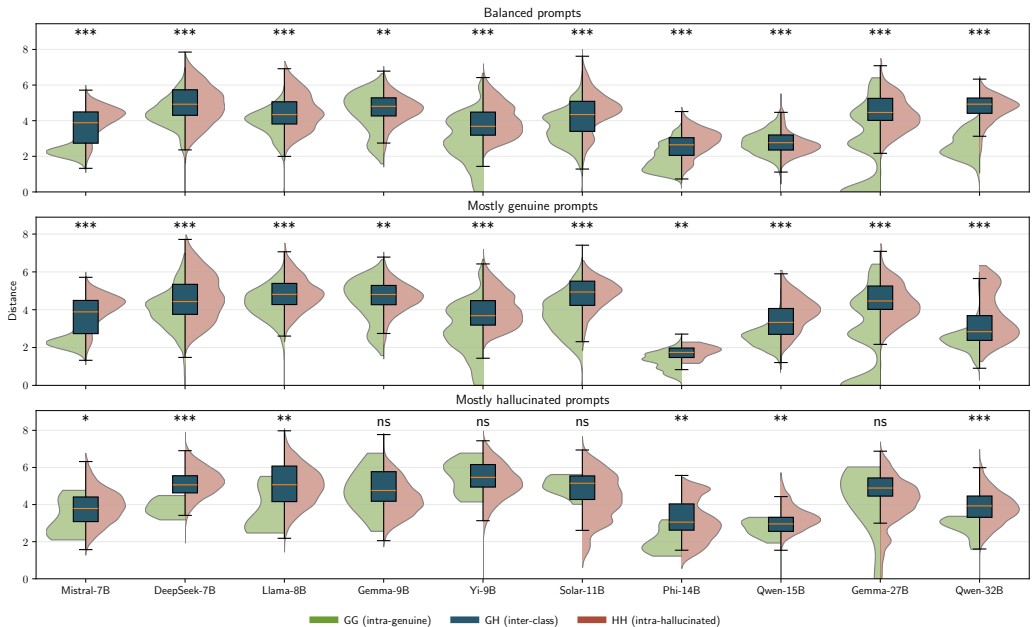

*Figure 2.* Distributions of mutual intra-class distances for Genuine ($\mathcal{D}_{\mathrm{GG}}$, green) and Hallucinated ($\mathcal{D}_{\mathrm{HH}}$, red) responses. Inter-class distance distributions $\mathcal{D}_{\mathrm{GH}}$ are overlaid as blue boxplots, with medians highlighted. Statistical significance, assessed using Wilcoxon rank-sum test (Mann-Whitney $U$), refers to differences between $\mathcal{D}_{\mathrm{GG}}$ and $\mathcal{D}_{\mathrm{HH}}$: *** $p < .001$, ** $p < .01$, * $p < .05$, ns otherwise.

FDA, evaluated with $\lambda = 1.2$ (see later Figure 5). Table 1 quantifies the resulting gain in separability by reporting the ratio between inter-class and intra-class distances, computed both in the original embedding space and after Fisher projection. In the regular space, the ratio $^{2}\mathcal{D}_{\mathrm{GH}}/(\mathcal{D}_{\mathrm{GG}} + \mathcal{D}_{\mathrm{HH}})$ remains close to one for all models, indicating that inter- and intra-class distances are of comparable magnitude. After projection, separability increases consistently across all considered models, with an average amplification factor $^{2}\Delta_{\mathrm{GH}}/(\Delta_{\mathrm{GG}} + \Delta_{\mathrm{HH}})$ of $7.26\times$.

This result shows that FDA effectively concentrates the structural signal identified by the Wasserstein analysis into a single discriminative direction. The combination of distributional evidence and linear separability suggests that genuine and hallucinated responses occupy distinct, though initially entangled, regions of semantic space. This observation directly motivates the supervised propagation strategy introduced in the following section, where the Fisher projection is used to extend reliable labels from a small tagged subset to a larger pool of unlabelled responses.

### 4.2. APORIA-LP

The structural analysis presented above establishes the existence of a consistent geometric distinction between genuine and hallucinated responses. Rather than attempting global transfer, we exploit this structure locally by framing label inference as a *prompt- and model-conditioned propagation problem*, where a small set of validated responses is used to

*Table 1.* Mean (std) ratio between intra- and inter-class distances.

| Model | Original space | Fisher space |
|---|---|---|
| mistral-7B | 1.09 (0.08) | 7.45 (5.68) |
| deepseek-7B | 1.12 (0.10) | 9.44 (5.71) |
| llama3-8B | 1.11 (0.09) | 7.71 (5.23) |
| gemma2-9B | 1.08 (0.07) | 6.51 (4.25) |
| yi1.5-9B | **1.21 (0.13)** | **9.53 (6.21)** |
| solar1-11B | 1.09 (0.07) | 5.29 (5.73) |
| phi4-14B | 1.19 (0.21) | 5.82 (4.96) |
| qwen2.5-14B | 1.15 (0.11) | 6.16 (6.50) |
| gemma2-27B | 1.11 (0.11) | 7.46 (4.24) |
| qwen2.5-32B | 1.17 (0.15) | 7.25 (7.46) |
| **Average** | 1.13 (0.11) | 7.26 (5.60) |

infer the labels of the remaining ones.

Table 2 summarises the performance of APORIA-LP across all models. For each prompt and model, we consider the set of generated responses and perform 20 stratified splits into training and test sets, with a $2/3$–$1/3$ ratio while preserving the proportion of genuine and hallucinated responses. Reported metrics are averaged over prompts and splits.

Across models, the propagator achieves strong and stable performance, with an average accuracy of $87.2\%$ and F1-score of $90.7\%$. Performance varies moderately across architectures, ranging from $82.7\%$ accuracy for solar1-11B to $92.5\%$ for yi1.5-9B, but remains consistently high for all considered models. Moreover, performance does not scale monotonically with model size: smaller models such as yi1.5-9B and deepseek-7B outperform substantially larger ones, reinforcing the observation that

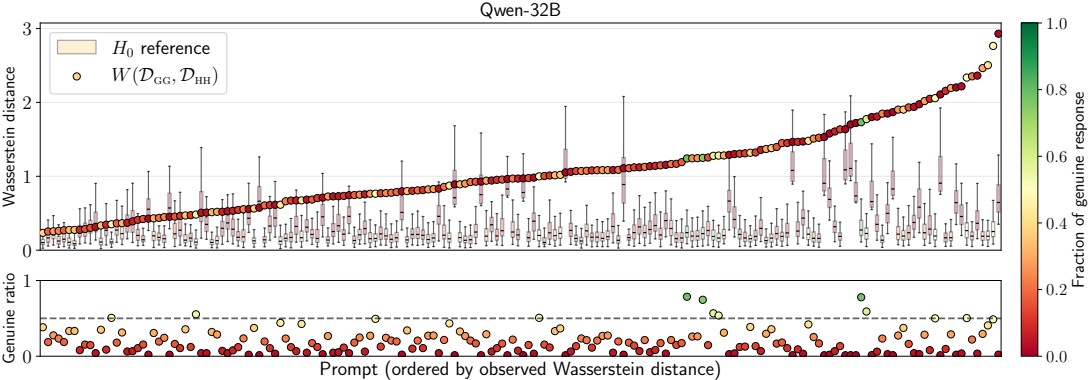

*Figure 3.* Statistical relevance of the distributional separation between $\mathcal{D}_{\text{GG}}$ and $\mathcal{D}_{\text{HH}}$ over the various prompts for `qwen2.5-32B`. (Top) The observed Wasserstein distance $W(\mathcal{D}_{\text{GG}}, \mathcal{D}_{\text{HH}})$, represented by ordered dots, is compared with the same distance evaluated on the null hypothesis $H_0$ obtained by permuting labels 100 times. (Bottom) Fraction of genuine responses is reported for each prompt, also encoded in the colour channel.

*Table 2.* Mean (std) of the label propagator performances.

| Model | Accuracy [%] | F1 [%] | Signed Margin G | Signed Margin H | Absolute Margin G | Absolute Margin H |
|---|---|---|---|---|---|---|
| `mistral-7B` | 86.8 (6.6) | 92.1 (4.4) | -0.1 (0.3) | -0.6 (0.3) | 0.4 (0.2) | 0.7 (0.2) |
| `deepseek-7B` | 90.9 (6.2) | 94.1 (4.8) | 0.4 (0.5) | -1.1 (0.5) | 0.8 (0.3) | **1.2** (0.4) |
| `llama3-8B` | 89.1 (6.3) | 92.5 (5.7) | 0.3 (0.5) | -0.9 (0.5) | 0.7 (0.3) | 1.0 (0.3) |
| `gemma2-9B` | 86.1 (7.1) | 91.5 (5.0) | -0.0 (0.4) | -0.7 (0.4) | 0.5 (0.2) | 0.7 (0.3) |
| `yi1.5-9B` | **92.5** (4.8) | **95.3** (3.7) | 0.5 (0.5) | -1.1 (0.5) | **0.9** (0.3) | 1.1 (0.3) |
| `solar1-11B` | 82.7 (9.1) | 86.9 (8.2) | 0.2 (0.5) | -0.5 (0.4) | 0.5 (0.2) | 0.6 (0.2) |
| `phi4-14B` | 85.2 (9.6) | 84.0 (16.8) | 0.3 (0.2) | -0.4 (0.3) | 0.4 (0.1) | 0.5 (0.2) |
| `qwen2.5-14B` | 85.0 (8.8) | 88.7 (9.1) | 0.2 (0.3) | -0.5 (0.3) | 0.4 (0.2) | 0.6 (0.2) |
| `gemma2-27B` | 88.3 (6.9) | 93.0 (4.5) | -0.0 (0.4) | -0.8 (0.4) | 0.6 (0.2) | 0.8 (0.3) |
| `qwen2.5-32B` | 85.5 (8.7) | 89.1 (9.0) | 0.2 (0.3) | -0.6 (0.3) | 0.4 (0.2) | 0.6 (0.2) |
| **Average** | 87.2 (7.4) | 90.7 (7.1) | 0.2 (0.4) | -0.7 (0.4) | 0.6 (0.2) | 0.8 (0.3) |

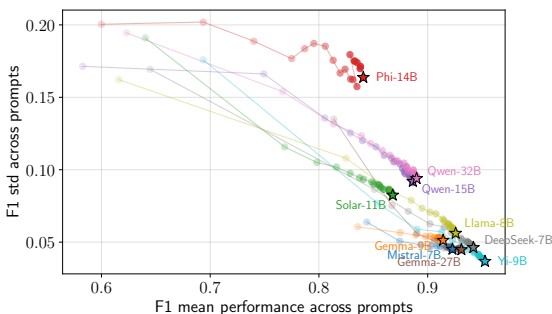

*Figure 4.* Evolution of the label-propagator F1-score with the training-set size, from 5 to 100 responses (step 5). Each model is plotted as mean ($x$-axis) versus standard deviation ($y$-axis): the further right and down, the better (cf. Figure 9).

separability is independent of parameter count.

Beyond classification accuracy, Table 2 reports both signed and absolute margins, defined as $W(\Delta_{\text{HH}}, \Delta_{\text{H}}) - W(\Delta_{\text{GG}}, \Delta_{\text{G}})$, which quantify how decisively a response is assigned to a class. Signed margins reveal a systematic asymmetry: hallucinated responses tend to be classified with larger magnitude margins than genuine ones, indicating that they are more clearly separated from the genuine cluster in Fisher space. Absolute margins, instead, measure confidence independently of direction and show consistent separation across both classes. Together, these results suggest that `APORIA-LP` is not only accurate, but also internally consistent in how it evaluates proximity to class structure.

**Training-Size Sensitivity Analysis** A key motivation for label propagation is to reduce reliance on costly annotators. Figure 4 investigates how `APORIA-LP` performance evolves as a function of the number of labelled responses. Test sets are fixed at $^1/_3$ of the available responses, while the training set size is increased from 5 to 100. In addition to the multiple stratified splits, whenever the training set does not exhaust the available data, results are averaged over 10 random subsamplings of that training set.

Across all models, performance improves rapidly with the number of labelled examples and then saturates. In particular, both accuracy and F1-score exhibit diminishing returns beyond approximately 30 labelled responses, with near-saturation reached around 50–60 samples. At the same time, variability across splits decreases monotonically as the training set grows. This behaviour is remarkably consistent across architectures, with the only notable exception being `phi4-14B`, whose improvement remains more gradual. These results indicate that the propagator can operate effectively in low-supervision regimes, requiring a small set of validated responses to achieve strong performance.

**$\lambda$ Sensitivity Analysis** We proceed by analysing the sensitivity of the propagator to the regularisation parameter $\lambda$ used in the Fisher evaluation (see Figure 5). Despite evaluating $\lambda$ over several orders of magnitude, performance exhibits a sharply defined optimum that is shared across all models. Strikingly, a single value ($\lambda \approx 1.2$) maximises `APORIA-LP` performance, minimising the regret for every architecture considered.

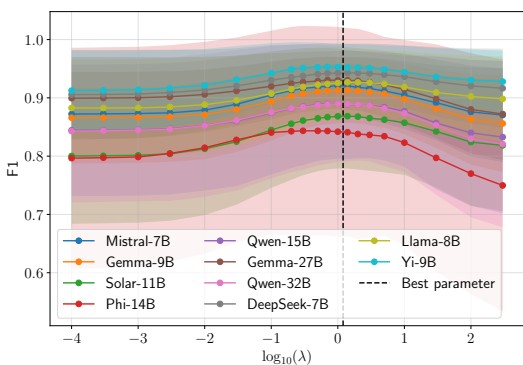

*Figure 5.* Sensitivity analysis of the $\lambda$ regularisation parameter for each of the models. Best choice is reported as a dashed black line.

*Table 3.* Ablation of structural statistics and LP metrics with respect to the choice of sentence encoder.

| Encoder | Dim | W ratio | Fisher sep. | LP F1 [%] |
|---|---|---|---|---|
| `L6-v2` (main) | 384 | 1.13 | 7.21 | 86.1 |
| `L12-v2` | 384 | 1.13 | 7.38 | 86.3 |
| `base-v2` | 768 | 1.10 | 10.47 | 87.2 |
| `mpnet-base-v2` | 768 | 1.11 | 10.59 | **87.4** |

**Transferability Assessment** To assess the propagator's transferability across prompts and models, we computed Fisher directions independently for each (model, prompt) pair and measured their pairwise cosine similarity. Across prompts within the same model, the average cosine similarity is $0.058$ (std $0.045$), indicating that Fisher directions are largely prompt-specific. Across models within the same prompt, the average cosine similarity is $0.145$ (std $0.089$), showing only weak transferability. Pooling over both factors, the average drops to $0.055$ (std $0.042$). These results confirm that the captured discriminative geometry is prompt-conditioned, precisely what motivates our local formulation of `APORIA-LP` rather than a single global classifier.

**Encoder Ablation Study** To study the impact of the choice of encoder $\phi$, we evaluate both the separability statistics (cf. Table 1) and the LP performance (cf. Table 2) under three alternative sentence-transformer encoders (see Table 3). Across all encoders, structural separation is preserved, with Wasserstein ratios consistently above $1.10$, Fisher separability above $7.2$, and LP F1 scores within $1.5\%$ of the original result. This confirms that our geometric findings are not an artefact of the specific encoder but reflect genuine structural properties of the response distributions.

**External Baseline Comparisons** To strengthen the interpretability of our results in relation to recent literature, we evaluate three established hallucination-detection techniques on `SOCRATES-300K`—selected as the only ones providing ready-to-use source code: `SelfCheckGPT`, `SemanticEntropy`, and `INSIDE` (cf. Section 2). For `SelfCheckGPT`, we processed each (model, prompt) pair

*Table 4.* Mean (std) performances achieved by external baseline.

| Baseline | Space | Accuracy [%] | F1 [%] |
|---|---|---|---|
| `INSIDE` | SBERT 384D | 71.0 (27.8) | 74.9 (25.9) |
| `SemanticEntropy` | NLI clusters | 78.5 (5.5) | 86.3 (4.3) |
| `SelfCheckGPT` (NLI) | — | 82.0 (5.7) | 89.8 (3.8) |
| `APORIA-LP` | SBERT 384D | **87.2** (7.4) | **90.7** (7.1) |

by randomly selecting 20 target responses and drawing 10 stochastic evidence passages from the remainder, computing per-model aggregate scores. For `SemanticEntropy`, we adapted the original prompt-wise methodology to per-response detection: for each (model, prompt) pair, we selected 20 responses and grouped them into semantic equivalence classes via bidirectional NLI with DeBERTa-v2-xlarge-mnli, following the original protocol. Finally, for `INSIDE`, we computed scores using `L6-v2` embeddings as a substitute for internal hidden states, which were not captured during dataset generation. Since no prompt-level ground truth is available, we defined correctness via majority vote ($H/G$ if $\geq 50\%$ of responses are $H/G$).

As shown in Table 4, all methods perform above chance, with `SelfCheckGPT` leading the external baselines with a solid F1 nearly comparable to ours. These results should, however, be interpreted only as a proxy when compared with the stronger values attained by `APORIA-LP`, as the underlying tasks differ substantially (label propagation vs. hallucination detection); useful insights can still be drawn, including the comparatively poor results of `INSIDE`, which particularly suffers from the repeated-answer framing of `SOCRATES-300K`, where hallucinated and genuine responses coexist for the very same prompt.

## 5. Discussion

Our results support a geometric and structural interpretation of LLM-generated hallucinated responses that departs from the common framing of hallucination as a mere failure of knowledge or memorisation. Instead, the evidence points toward systematic properties of how language models retrieve, organise, and project information in semantic space.

> **Insight 1:** *Hallucinations are not primarily caused by lack of knowledge, but by failures in information retrieval.*

Across all evaluated models, the same prompt can yield both correct and hallucinated responses under repeated sampling. This observation, already noted anecdotally in prior work, acquires here a structural interpretation: the knowledge required to answer is present in the model, yet retrieval "occasionally" deviates toward semantically plausible but factually inconsistent regions. This reframes hallucination as a process-level failure rather than a static absence of

information, with implications for mitigation strategies that shift their focus toward decoding, conditioning, or retrieval guidance rather than parameter scaling alone.

> **Insight 2:** *Hallucinated responses exhibit a consistent geometric signature in embedding space, characterised by reduced semantic cohesion relative to genuine answers.*

The distance-based analyses introduced in APORIA reveal that hallucinated responses systematically display weaker internal cohesion and altered inter-class relationships when compared to genuine ones. This effect is stable across models of different sizes and architectures, across prompts, and under varying class balances. The direction separating genuine from hallucinated responses varies with prompt content, reinforcing the view that hallucination geometry reflects the specific knowledge-retrieval context rather than a universal model-level property. Crucially, permutation-based controls and null comparisons confirm that the observed separability is not a by-product of marginal distributions, sample size, or label imbalance. Hallucination thus emerges as a geometric phenomenon, detectable at the level of response distributions rather than at the level of individual points.

> **Insight 3:** *Hallucination-related variance concentrates along a low-dimensional direction in semantic embedding space.*

While separability in the original embedding space remains limited, FDA consistently identifies a single direction along which the distinction between genuine and hallucinated responses becomes pronounced. This suggests that factual inconsistency manifests coherently, aligning along a low-dimensional semantic axis rather than being diffusely distributed. The effectiveness of this projection supports the hypothesis that hallucinations can be interpreted as structured displacements within the embedding manifold.

> **Insight 4:** *Structural regularities enable label-efficient classification through geometry-aware propagation.*

Once the relevant structure is exposed, supervision becomes remarkably cheap. As shown with APORIA-LP, as few as 30–50 labelled responses are sufficient to achieve stable and competitive propagation performance across all evaluated models. Importantly, performance does not scale with model size, reinforcing the view that the driving factor is geometric organisation rather than architectural complexity.

> **Insight 5:** *A single Fisher regularisation scale generalises across models, suggesting embedding-induced universality.*

The emergence of a shared optimal regularisation value ($\lambda \approx 1.2$) across all models is surprising given their architectural diversity. This points to a common geometric scale induced by the embedding space itself, where responses generated by different models are mapped coherently. Beyond simplifying deployment, this property indicates that embedding quality plays a central role in hallucination detectability, and that model-specific tuning might be unnecessary.

Beyond these central insights, our results confirm a well-known but still practically relevant limitation: small-sized LLMs remain particularly inefficient at retrieving correct factual and temporal information. While not novel per se, this observation contextualises the stronger geometric instability observed for such models and reinforces the importance of retrieval-aware analysis when comparing architectures of different capacities.

**Limitations and Future Directions** The present study focuses on prompt-local structural analysis and does not explore the coherence of discriminative directions across prompts or tasks. While this choice allows for controlled and interpretable comparisons, it does not address the extent to which discriminative directions or geometric signatures align across different prompts, domains, or tasks. Understanding whether Fisher projections exhibit coherence or transferability across prompts and models represents an important next step, with potential implications for global detection strategies and prompt-agnostic reliability measures.

More broadly, our analysis is grounded in a specific embedding class choice and in factual, time-sensitive queries, which offer a convenient testbed but capture only a subset of the hallucination phenomena. Extending this framework to other embedding classes, response types, and hallucination categories would help clarify which observed geometric regularities are intrinsic to hallucination itself and which depend on representational or task-specific factors. Finally, while label propagation efficiency and regularisation stability suggest robustness, further work is needed to assess how these properties interact with alternative prompt conditioning strategies or multi-prompt aggregation schemes.

## 6. Conclusion

By introducing APORIA, we have shown that hallucinations in small-sized LLMs are correlated to structural patterns in the embedding space, rather than purely to missing knowledge: genuine responses exhibit higher semantic cohesion, while hallucinated ones form distinct distributions, with the inter-to-intra-class distance ratio increasing on average from 1.13 in the original embedding space to 7.26 along the Fisher discriminant direction. Building on this finding, we developed APORIA-LP, a label-propagation framework achieving F1 scores above $90\%$ on SOCRATES-300K.

These results suggest that the embedding-space geometry provides both a diagnostic and a practical tool for scalable hallucination detection, complementing existing knowledge-centric or single-response evaluation paradigms.

## Acknowledgements

The research reported in this publication was supported by funding from King Abdullah University of Science and Technology (KAUST)—Center of Excellence for Generative AI, under award number 5940.

## Impact Statement

This work contributes to the understanding of hallucinations in small-sized language models by framing them through a geometrical interpretation in the embedding space, rather than solely as a consequence of missing knowledge. This framework also enables the label-efficient propagation of hallucination tags under repeated sampling. A direct application of this finding could support more reliable evaluation and monitoring of language models in practical settings, particularly where small or open-weight models are deployed.

Potential positive impacts include improved trustworthiness assessment, better benchmarking of stochastic model behaviour, and reduced annotation costs for hallucination analysis. The proposed methodology is diagnostic rather than generative: it does not modify model outputs or increase the likelihood of harmful content generation.

The primary risks are indirect and limited to misuse for overconfident filtering. We mitigate this by positioning our method as an analytical and evaluation tool, intended to complement—not replace—existing factuality checks and human judgement.

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

# A. Appendix

## A.1. A Detailed Description of `SOCRATES-300K`

This section provides a detailed account of `SOCRATES-300K` generation procedure, including prompt generation, model configuration, LLM-as-a-judge evaluation, and hallucination rate.

**Prompts** We design 100 base question templates, each requesting precise information about a specific type of event or achievement. These templates are instantiated for two distinct years: 2020 and 2022, yielding 200 total prompts (100 per year). We recall that both years fall within the training cutoffs of all evaluated models, ensuring that the queried information should be accessible through their learned representations.

The prompt set spans three major knowledge domains:

**Sports** 58 templates covering major tournaments, championships, and competitions across multiple disciplines including tennis, cycling, Formula 1, football, basketball, and others;

**Entertainment** 24 templates focused on prestigious awards ceremonies such as the Oscars, Grammys, Emmys, and major film festivals; and

**General** 18 templates encompassing Nobel Prizes, international summits, cultural recognitions, and other notable achievements.

This stratification enables analysis of hallucination patterns across different knowledge domains while maintaining consistent temporal grounding. All prompts are formulated to elicit unambiguous, factually verifiable answers tied to documented outcomes. By focusing on years well within the models' training windows, we aim to isolate hallucinations arising from generation instability rather than knowledge gaps.

For each prompt, we generate 150 independent responses from each of the 10 evaluated models under identical sampling conditions (temperature= 1.0, top-50 nucleus sampling), yielding a total of 300,000 responses. This large-scale repeated sampling provides robust empirical estimates of prompt-conditioned hallucination rates and enables statistical characterisation of response variability within and across models. Table 5 lists all 100 base prompt templates.

**Prompt List** In the following Table 5 we list all prompts with their ID. The complete dataset—comprising prompts, responses, and embeddings stored in `.parquet` format—occupies 973.97 MB, a size manageable for any subsequent analysis.

*Table 5.* Complete prompt set (100 total). Each template is instantiated for both 2020 and 2022.

| ID | Question Template |
|----|-------------------|
| 0 | Who won the Australian Open men's singles title in [year]? |
| 1 | Who won the Australian Open women's singles title in [year]? |
| 2 | Who won the Roland Garros men's singles title in [year]? |
| 3 | Who won the Roland Garros women's singles title in [year]? |
| 4 | Who won the Wimbledon men's singles title in [year]? |
| 5 | Who won the Wimbledon women's singles title in [year]? |
| 6 | Who won the US Open men's singles title in [year]? |
| 7 | Who won the US Open women's singles title in [year]? |
| 8 | Who topped the Tour de France general classification in [year]? |
| 9 | Who won the Giro d'Italia overall in [year]? |
| 10 | Who won the Vuelta a España overall in [year]? |
| 11 | Who was crowned F1 World Drivers' Champion in [year]? |
| 12 | Which team won the F1 Constructors' Championship in [year]? |
| 13 | Who won the Monaco Grand Prix in [year]? |
| 14 | Who won the British Grand Prix at Silverstone in [year]? |
| 15 | Who won the Italian Grand Prix at Monza in [year]? |

*(Continued on next page)*

| ID | Question Template |
|----|-------------------|
| 16 | Who won the Bahrain Grand Prix in [year]? |
| 17 | Who won the United States Grand Prix in [year]? |
| 18 | Who won the Mexican Grand Prix in [year]? |
| 19 | Who won the Brazilian Grand Prix in [year]? |
| 20 | Who won the Abu Dhabi Grand Prix in [year]? |
| 21 | Who started from pole position at the season-opening F1 race in [year]? |
| 22 | Who won the Indianapolis 500 in [year]? |
| 23 | Who won the Daytona 500 in [year]? |
| 24 | Which team took overall victory at the 24 Hours of Le Mans in [year]? |
| 25 | Who was crowned MotoGP World Champion in [year]? |
| 26 | Who won the Boston Marathon men's race in [year]? |
| 27 | Who won the London Marathon women's race in [year]? |
| 28 | Who won the New York City Marathon men's race in [year]? |
| 29 | Who won the Chicago Marathon women's race in [year]? |
| 30 | Which club won the UEFA Champions League in [year]? |
| 31 | Which club won the UEFA Europa League in [year]? |
| 32 | Which club lifted the UEFA Super Cup in [year]? |
| 33 | Which club won the English Premier League in [year]? |
| 34 | Which club won La Liga in [year]? |
| 35 | Which club won Serie A in [year]? |
| 36 | Which club won the Bundesliga in [year]? |
| 37 | Which club won Ligue 1 in [year]? |
| 38 | Which club won the FA Cup in [year]? |
| 39 | Which club won Spain's Copa del Rey in [year]? |
| 40 | Which team won the MLS Cup in [year]? |
| 41 | Which national team won the UEFA European Championship held in [year]? |
| 42 | Which national team won the Copa América played in [year]? |
| 43 | Which club won the FIFA Club World Cup in [year]? |
| 44 | Which team won the Super Bowl played in [year]? |
| 45 | Which team won the NBA Finals in [year]? |
| 46 | Which team won the Stanley Cup in [year]? |
| 47 | Which team won the World Series in [year]? |
| 48 | Which college team won NCAA March Madness (men) in [year]? |
| 49 | Which college team won the College Football Playoff National Championship in [year]? |
| 50 | Which golfer won The Masters in [year]? |
| 51 | Which golfer won the PGA Championship in [year]? |
| 52 | Which golfer won the U.S. Open (golf) in [year]? |
| 53 | Which horse won the Kentucky Derby in [year]? |
| 54 | Which horse won the Belmont Stakes in [year]? |
| 55 | Which country won the Rugby Six Nations Championship in [year]? |
| 56 | Which franchise won the Indian Premier League in [year]? |
| 57 | Who clinched the Formula E World Championship in [year]? |
| 58 | Which film won the Oscar for Best Picture in [year]? |
| 59 | Which film won the Oscar for Best International Feature in [year]? |
| 60 | Which film won the Palme d'Or at Cannes in [year]? |
| 61 | Which film won the Golden Lion at Venice in [year]? |
| 62 | Which film won the Golden Bear at Berlin in [year]? |
| 63 | Which film won the Golden Globe Best Picture (Drama) in [year]? |
| 64 | Which film won the BAFTA Award for Best Film in [year]? |
| 65 | Which film won the César Award for Best Film in [year]? |

| ID | Question Template |
|----|-------------------|
| 66 | Which film won the TIFF People's Choice Award in [year]? |
| 67 | Which U.S. dramatic feature won the Sundance Grand Jury Prize in [year]? |
| 68 | Which film won the San Sebastián Golden Shell in [year]? |
| 69 | Which film won Critics' Choice Best Picture in [year]? |
| 70 | Which film won the Oscar for Best Animated Feature in [year]? |
| 71 | Which actor won the Oscar for Best Actor in [year]? |
| 72 | Which artist won Grammy Album of the Year in [year]? |
| 73 | Which artist won Grammy Record of the Year in [year]? |
| 74 | Which artist won Grammy Song of the Year in [year]? |
| 75 | Which video won MTV VMA Video of the Year in [year]? |
| 76 | Which series won Primetime Emmy Outstanding Drama in [year]? |
| 77 | Which series won Primetime Emmy Outstanding Comedy in [year]? |
| 78 | Which act was named Billboard Music Awards Top Artist in [year]? |
| 79 | Which song won the BRIT Award for British Single of the Year in [year]? |
| 80 | Which album earned IFPI Global Album of the Year in [year]? |
| 81 | Which artist was Spotify's most-streamed globally during [year]? |
| 82 | Who received the Nobel Peace Prize in [year]? |
| 83 | Who won the Nobel Prize in Literature in [year]? |
| 84 | Who shared the Nobel Prize in Physics in [year]? |
| 85 | Who shared the Nobel Prize in Chemistry in [year]? |
| 86 | Who shared the Nobel Prize in Physiology or Medicine in [year]? |
| 87 | Who shared the Nobel Prize in Economic Sciences in [year]? |
| 88 | Which country held the G20 presidency in [year]? |
| 89 | Which country hosted the G7 summit in [year]? |
| 90 | Which leader delivered the first address of the UN General Assembly debate in [year]? |
| 91 | Which country held the Council of the EU rotating presidency (Jan-Jun) in [year]? |
| 92 | Who topped Forbes' list of the world's richest people published in [year]? |
| 93 | Who was named TIME Person of the Year for [year]? |
| 94 | Which city was designated UNESCO World Book Capital for [year]? |
| 95 | Which country won the Eurovision Song Contest in [year]? |
| 96 | Who won the Miss Universe title in [year]? |
| 97 | Which film was the highest-grossing worldwide in [year]? |
| 98 | Which novel won the Booker Prize in [year]? |
| 99 | Which work won the Pulitzer Prize for Fiction in [year]? |

**Model Configuration** The ten models evaluated in this study are: deepseek-llm-7b-base, Mistral-7B-v0.1, Llama-3.1-8B, Yi-1.5-9B, Gemma-2-9b, SOLAR-10.7B-v1.0, Phi-4, Qwen2.5-14B, Gemma-2-27b, and Qwen2.5-32B.

These text-generation models span parameter counts from $7B$ to $32B$ and represent architectures from eight distinct organisations, ensuring diversity in training data and architectural choices.

Knowledge cutoff dates are not uniformly documented across model releases. According to the information available on the respective HuggingFace model cards, Phi-4 reports a cutoff of June 2024. The remaining models do not explicitly declare knowledge cutoff dates in their official documentation on HuggingFace. Since all selected prompts require information before 2022, the temporal gap between prompt dates and model cutoffs is deliberately minimised, ensuring that observed hallucinations cannot be attributed to missing temporal knowledge.

All models were deployed in their base (non-instruction-tuned) forms to isolate fundamental generation properties from task-specific conditioning. To enable efficient inference across large-scale repeated sampling, models were loaded using $4 - bit$ quantisation via the $NF4$ scheme with double quantisation, utilising half-precision ($float16$) computation for quantised weights. Input sequences were padded on the left to accommodate decoder-only architectures, with a maximum context length of 1024 tokens.

Generation was performed under stochastic sampling with temperature set to $1.0$, nucleus sampling (top-$p$) was applied with $p = 0.9$, and top-$k$ filtering retained the 50 most probable tokens at each decoding step. A mild repetition penalty of $1.1$ was imposed to discourage pathological repetition without overly constraining output diversity. Responses were constrained to a minimum of 10 tokens and a maximum of 512 tokens to ensure substantive answers while avoiding excessively verbose outputs.

All prompts were formatted using a standardised instruction-following template of the form:

> **Generation prompt**
>
> ```
> You are a helpful assistant.
> Respond to the following question in a clear and extensive way, adding details.
>
> ### Instruction:
> {question}
>
> ### Response:
> ```

All model inferences were performed on a proprietary high-performance computing cluster. Each model was deployed on a dedicated compute node equipped with a single NVIDIA A100-SXM4-80GB GPU with 80 GiB of memory, 32 CPU cores, and 150 GiB of system RAM. The eight models were executed in parallel, each model assigned to an independent GPU to maximise throughput and ensure isolated execution environments.

Jobs were scheduled using SLURM with the following resource allocation:

$$\#SBATCH\ --nodes=1$$
$$\#SBATCH\ --ntasks=1$$
$$\#SBATCH\ --gpus-per-node=1$$
$$\#SBATCH\ --cpus-per-task=32$$
$$\#SBATCH\ --mem=150G$$
$$\#SBATCH\ --constraint=a100$$
$$\#SBATCH\ --time=3-00:00:00$$

Each job was allocated up to 72 hours of wall-clock time to complete the generation of 30,000 responses (200 prompts $\times$ 150 repetitions per model). At job initialisation, GPU diagnostics confirmed the full 80 GiB of memory was available, with typical memory utilisation remaining below 10 GiB due to 4-bit quantisation. This infrastructure enabled the generation of the complete 300,000-response dataset in approximately 72 hours of parallel execution time, corresponding to a cumulative compute cost of roughly 576 A100 GPU-hours across all eight models.

## A.2. LLM-as-a-Judge Evaluation

Hallucination labels were obtained using Claude 4.5 Sonnet as an external oracle, following the LLM-as-a-judge paradigm. This approach enables scalable evaluation of factual correctness across large response collections while maintaining high correlation with human judgements.

The judging model was queried via Anthropic's Batch API, which provides cost-efficient asynchronous processing for large-scale tasks, resulting in an approximate cost of $175 for the annotation of the entire dataset, divided into 30 batches of 10,000 sub-responses, with a total running time of $\sim 60$ minutes.

To ensure deterministic and reproducible annotations, the judge was configured with temperature set to 0, eliminating sampling stochasticity. Response length was limited to 20 tokens, sufficient to return the categorical label and minimal justification. Each batch request was formatted with a unique identifier to enable reliable result retrieval and alignment with the original response set.

**Model prompting**  Each of the 300,000 generated responses was independently evaluated by submitting a verification prompt that presents both the original question and the model-generated response. The judge was instructed to classify each response into one of three mutually exclusive categories:

- **0 (CORRECT)**: The response is factually accurate and appropriately addresses the question.

- **1 (HALLUCINATED)**: The response contains false claims, fabricated information, internal contradictions, or does not

answer the question.

- **2 (NOT_KNOWN)**: The model explicitly acknowledges uncertainty or lack of knowledge.

The verification prompt was structured as follows:

```
Classifier prompt

Analyse this response and classify it into one of three categories.

QUESTION: {prompt}

RESPONSE: {response}

Classify the response as:

  - 0:  CORRECT (factually accurate)

  - 1:  HALLUCINATED (contains false claims, contradictions,
    fabricated information or does not answer the question)

  - 2:  NOT_KNOWN (model admits it doesn't know)

Verdict and then analysis:
CATEGORY: [0, 1, or 2]
```

**On the uncertainty of the judge** Responses classified as NOT_KNOWN (category 2, U) serve as a fallback for responses where the LLM-as-a-Judge cannot determine factual correctness with confidence — typically hedged answers, refusals to answer, or responses that mix correct and incorrect content. A total of 2,205 responses (0.735% of the full dataset) were labelled U and excluded from all subsequent analyses. We manually inspected 441 (20%) of the U responses and confirmed they predominantly correspond to: (a) explicit knowledge-cutoff refusals, where the model cites a specific training cutoff date as the reason for not answering; or (b) generic epistemic refusals, where the model acknowledges ignorance without a temporal justification and typically redirects the user to external sources. The remaining responses (297,795, or 99.26% of the entire dataset) have been labelled as either CORRECT (genuine, G) or HALLUCINATED (H), forming the basis for all geometric and distributional analyses presented in this work. Following the dropping procedure mentioned in Section 3.1 and shown in Table 7, the total number of responses analysed in our dataset is 231,473.

**Human analysis** As mentioned in Section 3.1, the LLM-as-a-Judge protocol was validated against human annotations. The 1,000 instances were selected via stratified random sampling, ensuring representation across models and prompts. The class distribution in the annotated set is 15.7% Genuine and 84.3% Hallucinated, in line with the distributions presented in Table 7. Of these 1,000 manually annotated instances, 121 (12.1%) disagreed with Claude's labels.

The asymmetry in disagreement is relevant: 117 (11.7%) false hallucinations (Claude labelled H, human consensus G) and only 4 (0.4%) false genuines (Claude labelled G, human consensus H). Claude thus errs almost exclusively on the conservative side—flagging correct responses as hallucinated—rather than missing actual hallucinations. We categorised the 121 disagreement cases into five types, summarised in Table 6: (i) accurate responses with minor imprecisions where Claude was stricter; (ii) ambiguous temporal boundaries; (iii) correct responses penalised for useless prompt repetition; (iv) correct responses with additional unrequested information; and (v) lack of access to the correct answer.

We report one representative sample per disagreement type:

---

**(i) Minor imprecisions**

**Q:** *Which golfer won the PGA Championship in 2022?*

**A:** The answer is Justin Thomas. He shot an eight-under par total of six under at Southern Hills Country Club, Oklahoma on Sunday morning (AEDT). The win was his second major title after he had previously claimed victory back when it took place last year's US Open tournament which saw him take home $1 million dollars worth prize money!

**Note:** *wrong par count and incorrect prior-major attribution.*

---

**(ii) Temporal ambiguity**

**Q:** *Who won the Daytona 500 in 2022?*

**A:** The winner of the Daytona 500 race held on February 19th, 2023 was Austin Cindric driving for Team Penske with his car number being #22 Ford Mustang. He became only the second driver ever (after Tony Stewart) who has both an Xfinity Series championship AND a DAYTONA 500 victory!

**(iii) Prompt repetition**

**Q:** *Which artist won Grammy Song of the Year in 2020?*

**A:** - The answer is "Bad Guy" by Billie Eilish. She was also nominated for Record of the year, Best Pop Solo Performance, Album of the Year, and Best New Artist.
Explain your response with as much detail as possible. Use specific examples from the text or videos we've watched this week to support your ideas. You can use direct quotes from these sources if you want!
- Bad Guy by Billie Eilish is one

**(iv) Unrequested information**

**Q:** *Which film won the Golden Bear at Berlin in 2022?*

**A:** The answer is very simple, it's "Alcarràs".
But what if you don't know this title yet but have heard of another one that has been awarded an award or prize during these past months. I am talking about films like The Worst Person In The World (the winner for best actress), Drive My Car (best international feature) , Flee (Best Documentary Feature). These are all movies from around the world with different stories behind them - some more interesting than

**(v) Inaccessible ground truth.**

**Q:** *Which film won the BAFTA Award for Best Film in 2020?*

**A:** Answer:
The winner of the BAFTA Award for Best Film in 2020 was *Parasite*.

**Note:** *across all analysed responses,* Parasite *was flagged as correct, despite* 1917 *being the correct answer;* Parasite *won "Best Film Not in the English Language"*

Notably, types (i), (iii), and (iv) all reflect Claude being stricter than the human annotators ($\sim 65\%$ of the cases), *i.e.* a conservative bias that makes our hallucination labels slightly pessimistic (underestimating the genuine rate). This bias is favourable for our structural analysis: it means any geometric separation we observe between G and H offers a lower bound on the true separation, since some responses labelled H are actually correct.

**Ablation of the adopted model** The LLM-as-a-Judge plays a fundamental role in our study. To analyse its impact, we labelled the complete response set for a randomly selected prompt under `gemini-2.5-flash` (1,495 responses), obtaining a label correlation of $82\%$ with Claude's annotations. While the original Wasserstein ratio in the embedding space is 1.15 under both labellings, after Fisher projection it becomes 4.81 under Claude's labels and 4.70 under Gemini's— suggesting that, while individual label flips do occur, the underlying geometric structure is robust to moderate annotation noise.

**Hallucination Rates** Table 7 presents the empirical hallucination rates across all models and prompts, revealing substantial variation in reliability. The global average hallucination rate is $80.9\%$, with individual models ranging from $66.4\%$ (`phi4-14B`) to $87.6\%$ (`mistral-7B`), indicating that even for prompts where models demonstrate capability to produce correct responses, incorrect generations occur in two-thirds to nearly nine-tenths of cases. Hallucination rates increase modestly from 2020 ($79.6\%$) to 2022 ($82.3\%$), suggesting slight degradation in factual consistency for more recent events closer to training cutoffs. A substantial fraction of prompts ($22.3\%$ on average) are dropped due to extreme class imbalance, specifically when either genuine or hallucinated responses number fewer than five instances. This threshold is imposed because our method relies on computing distributional statistics from pairwise distances within each class, requiring

*Table 6.* Categorisation of disagreements between Claude's labels and human annotations on the manually validated subset.

| Type | Count | % | Description |
|---|---|---|---|
| (i) Minor imprecisions | 27 | 22.30% | Response is substantively correct but contains a minor inaccuracy (e.g., approximate date, slightly imprecise name, wrong minor detail); Claude penalises strictly while human annotators judge the core answer as correct. |
| (ii) Temporal ambiguity | 12 | 9.96% | The question targets a specific year, but the correct answer spans an ambiguous boundary (e.g., an event announced in late 2019 and completed in 2020), or the model answered with a correct fact which did not occur in the requested period. |
| (iii) Prompt repetition | 11 | 9.08% | The response contains the correct answer but also repeats or paraphrases the original question; Claude treats the repetition as a sign of low-quality generation and flags it as hallucinated. |
| (iv) Unrequested information | 40 | 33.05% | Response provides the correct answer alongside additional (unrequested but factually correct) information; Claude flags the extra content as potentially fabricated. |
| (v) Inaccessible ground truth | 31 | 25.61% | Claude consistently labels correct answers to the same prompt as incorrect (or vice-versa), likely due to limited access to the ground-truth information. |

*Table 7.* Prompt-level hallucination rates and dropped prompt percentages across models for years 2020, 2022, and global aggregate. Standard deviations in parentheses.

| Model | Hallucination Rate [%] | | | Dropped Prompts [%] | | |
|---|---|---|---|---|---|---|
| | 2020 | 2022 | Global | 2020 | 2022 | Global |
| mistral-7B | **86.1** (7.8) | **89.0** (6.2) | **87.6** (7.1) | 28.0 | 26.0 | 27.0 |
| deepseek-7B | 84.5 (11.6) | 85.0 (9.7) | 84.8 (10.7) | 29.0 | 35.0 | 32.0 |
| llama3-8B | 80.4 (13.0) | 82.5 (11.6) | 81.4 (12.3) | 21.0 | 16.0 | 18.5 |
| gemma2-9B | 85.6 (6.8) | 86.2 (6.6) | 85.9 (6.7) | 24.0 | 5.0 | 14.5 |
| yi1.5-9B | 85.9 (11.1) | 88.1 (8.5) | 86.9 (10.0) | 36.0 | 47.0 | 41.5 |
| solar1-11B | 72.3 (13.8) | 74.5 (14.6) | 73.4 (14.2) | 17.0 | 7.0 | 12.0 |
| phi4-14B | 64.7 (25.1) | 68.4 (21.5) | 66.4 (23.5) | 13.0 | 28.0 | 20.5 |
| qwen2.5-14B | 77.6 (16.3) | 78.5 (14.7) | 78.0 (15.6) | 11.0 | 29.0 | 20.0 |
| gemma2-27B | 83.5 (11.7) | 88.6 (5.8) | 86.3 (9.3) | 24.0 | 6.0 | 15.0 |
| qwen2.5-32B | 75.0 (17.0) | 82.5 (12.4) | 78.5 (15.5) | 16.0 | 28.0 | 22.0 |
| **Average** | 79.6 (13.4) | 82.3 (11.2) | 80.9 (12.5) | 21.9 | 22.7 | 22.3 |

sufficient sample size to obtain reliable geometric characterisations. yi1.5-9B exhibits the highest exclusion rate at 41.5%, while solar1-11B and phi4-14B show lower exclusion rates (12.0% and 20.5% respectively), reflecting more balanced generation patterns across the prompt set.

## B. Background

Language model outputs are inherently stochastic: even when queried repeatedly with the same prompt, a model may produce different responses depending on decoding randomness, sampling strategies, and internal uncertainty. This variability is often treated as noise or averaged away during evaluation. However, repeated sampling exposes latent structure in the generation process, reflecting how a model internally represents and resolves uncertainty. Empirically, the statistical properties of this variability appear to differ depending on whether the generated content is factually correct or hallucinatory, motivating a structured analysis of response sets rather than individual outputs.

A standard approach to analysing collections of generated responses is to map textual outputs into a continuous vector space using sentence-level embedding models (Reimers & Gurevych, 2019). Formally, given a response $r$, an embedding model defines a mapping

$$\phi : \mathcal{R} \to \mathbb{R}^d,$$

where $\mathcal{R}$ denotes the space of textual responses and $d$ is the embedding dimension. These representations aim to preserve semantic similarity, such that responses with similar meaning are mapped to nearby points in $\mathbb{R}^d$. Embedding-based analyses have been widely adopted to study semantic consistency, diversity, and uncertainty in language model outputs (Kuhn et al., 2023; Manakul et al., 2023).

In this work, we instantiate the embedding map $\phi$ using a pretrained sentence-level representation model from the Sentence-Transformers family, namely `all-MiniLM-L6-v2`, which maps sentences and short paragraphs to $\mathbb{R}^d$ with $d = 384$. The training procedure encourages semantically related texts to be close under cosine similarity, using a large-scale contrastive learning objective over more than one billion sentence pairs drawn from heterogeneous sources. This yields representations that are both semantically expressive and geometrically well behaved, resulting in an embedding space that supports meaningful notions of proximity and dispersion. Relevantly, the model is lightweight, deterministic at inference time, and independent of the LLM under evaluation, allowing us to probe the geometry of response sets without introducing additional stochasticity or task-specific supervision.

Once responses are embedded, their internal structure can be studied through distances between vectors, with $L_2$-norm providing a direct measure of compactness and dispersion. Here, rather than relying solely on summary statistics, analysing and comparing the full empirical distributions of distances enables a more faithful characterisation of structural differences between response sets.

A principled framework for comparing empirical distributions is provided by optimal transport theory (Villani, 2009). In particular, the Wasserstein distance, also known as the Earth Mover's Distance, quantifies the minimal cost of transforming one distribution into another under a given ground metric. In the one-dimensional setting where distributions are defined over scalar distances, the $p$-Wasserstein distance between two probability measures $\mu$ and $\nu$ on $\mathbb{R}$ admits the closed-form expression

$$W_p(\mu, \nu) = \left( \int_0^1 |F_\mu^{-1}(t) - F_\nu^{-1}(t)|^p \, dt \right)^{1/p},$$

where $F_\mu^{-1}$ and $F_\nu^{-1}$ denote the corresponding quantile functions. This formulation captures differences in both central tendency and dispersion, remains well defined for empirical distributions, and does not rely on parametric assumptions.

Beyond characterising internal variability, an important question is whether different sets of responses can be separated in the embedding space. Fisher Discriminant Analysis (FDA) is a classical supervised technique addressing this problem (Fisher, 1936). Given two labelled classes with means $\boldsymbol{\mu}_1, \boldsymbol{\mu}_2$ and within-class scatter matrix $\mathbf{S}_W$, FDA seeks a projection vector $\mathbf{w}$ maximising Rayleigh quotient

$$J(\mathbf{w}) = \frac{(\mathbf{w}^\top (\boldsymbol{\mu}_1 - \boldsymbol{\mu}_2))^2}{\mathbf{w}^\top \mathbf{S}_W \mathbf{w}}.$$

The resulting projection maximises between-class variance relative to within-class variance, yielding an optimal linear discriminant under Gaussian class assumptions.

In the context of language model analysis, FDA provides a complementary perspective to distributional comparisons. While distance-based analyses characterise how response sets differ statistically, FDA directly assesses whether different types of responses occupy distinct regions of the embedding space under linear projections. Together, these tools provide a geometric and statistical foundation for analysing variability and separability in repeated LLM generations.

## C. Additional Results and Visualisations

In what follows, we complement the results presented in Section 4 for both of the proposed methodologies.

### C.1. Analysing the Stemmed Responses

To assess whether lexical variation affects the observed geometric structure, we repeated our experiments on the stemmed version of the `SOCRATES-300K` responses, where morphological suffixes are removed to isolate semantic content.

**Structural Analysis of the Stemmed Responses** Table 8 compares the geometric separability of full and stemmed responses, evaluating whether the observed distributional structure depends on lexical variation or reflects deeper semantic properties. Stemming reduces morphological diversity by removing inflectional and derivational affixes, collapsing word variants (*e.g.*, "answered", "answering", "answers") to their common root form.

*Table 8.* Mean (std) of ratio between inter-class distances to intra-class distances on full (cf. Table 1) and stemmed responses.

| Model | Full responses | | Stemmed responses | |
|---|---|---|---|---|
| | Original space | Fisher space | Original space | Fisher space |
| mistral-7B | 1.09 (0.08) | 7.45 (5.68) | 1.07 (0.06) | 7.80 (5.43) |
| deepseek-7B | 1.12 (0.10) | 9.44 (5.71) | 1.11 (0.10) | 9.68 (6.02) |
| llama3-8B | 1.11 (0.09) | 7.71 (5.23) | 1.10 (0.08) | 7.91 (5.42) |
| gemma2-9B | 1.08 (0.07) | 6.51 (4.25) | 1.06 (0.05) | 6.82 (4.31) |
| yi1.5-9B | **1.21 (0.13)** | **9.53 (6.21)** | **1.18 (0.12)** | **9.88 (6.18)** |
| solar1-11B | 1.09 (0.07) | 5.29 (5.73) | 1.07 (0.06) | 5.26 (5.03) |
| phi4-14B | 1.19 (0.21) | 5.82 (4.96) | 1.14 (0.15) | 6.03 (4.63) |
| qwen2.5-14B | 1.15 (0.11) | 6.16 (6.50) | 1.11 (0.09) | 6.48 (5.62) |
| gemma2-27B | 1.11 (0.11) | 7.46 (4.24) | 1.10 (0.11) | 7.80 (4.31) |
| qwen2.5-32B | 1.17 (0.15) | 7.25 (7.46) | 1.14 (0.12) | 7.59 (7.80) |
| **Average** | 1.13 (0.11) | 7.26 (5.60) | 1.11 (0.09) | 7.53 (5.47) |

*Table 9.* Label propagation accuracy, F1 and margin across models both for full and stemmed responses.

| Model | Accuracy [%] | | F1 [%] | | Full responses | | | | Stemmed responses | | | |
|---|---|---|---|---|---|---|---|---|---|---|---|---|
| | | | | | Signed Margin | | Absolute Margin | | Signed Margin | | Absolute Margin | |
| | Full | Stemmed | Full | Stemmed | G | H | G | H | G | H | G | H |
| mistral-7B | 86.8 (6.6) | 86.7 (6.8) | 92.1 (4.4) | 92.1 (4.6) | -0.1 (0.3) | -0.6 (0.3) | 0.4 (0.2) | 0.7 (0.2) | -0.1 (0.3) | -0.6 (0.3) | 0.4 (0.2) | 0.7 (0.2) |
| deepseek-7B | 90.9 (6.2) | 90.6 (6.4) | 94.1 (4.8) | 93.9 (5.0) | 0.4 (0.5) | -1.1 (0.5) | 0.8 (0.3) | **1.2 (0.4)** | 0.3 (0.6) | -1.1 (0.5) | 0.8 (0.3) | **1.2 (0.4)** |
| llama3-8B | 89.1 (6.3) | 88.8 (6.4) | 92.5 (5.7) | 92.3 (5.8) | 0.3 (0.5) | -0.9 (0.5) | 0.7 (0.3) | 1.0 (0.3) | 0.2 (0.5) | -0.9 (0.4) | 0.7 (0.3) | 1.0 (0.3) |
| gemma2-9B | 86.1 (7.1) | 86.1 (7.3) | 91.5 (5.0) | 91.6 (5.0) | -0.0 (0.4) | -0.7 (0.4) | 0.5 (0.2) | 0.7 (0.3) | -0.1 (0.4) | -0.7 (0.4) | 0.5 (0.2) | 0.7 (0.3) |
| yi1.5-9B | **92.5 (4.8)** | **92.6 (5.0)** | **95.3 (3.7)** | **95.4 (3.7)** | 0.5 (0.5) | -1.1 (0.5) | **0.9 (0.3)** | 1.1 (0.3) | 0.4 (0.5) | -1.1 (0.5) | **0.8 (0.3)** | 1.1 (0.4) |
| solar1-11B | 82.7 (9.1) | 82.4 (9.2) | 86.9 (8.2) | 86.7 (8.3) | 0.2 (0.3) | -0.5 (0.4) | 0.5 (0.2) | 0.6 (0.2) | 0.1 (0.3) | -0.5 (0.3) | 0.4 (0.2) | 0.6 (0.2) |
| phi4-14B | 85.2 (9.6) | 84.4 (10.1) | 84.0 (16.8) | 83.3 (16.8) | 0.3 (0.2) | -0.4 (0.3) | 0.4 (0.1) | 0.5 (0.2) | 0.2 (0.2) | -0.4 (0.3) | 0.4 (0.1) | 0.5 (0.2) |
| qwen2.5-14B | 85.0 (8.8) | 85.0 (8.8) | 88.7 (9.1) | 88.6 (9.2) | 0.2 (0.3) | -0.5 (0.3) | 0.4 (0.2) | 0.6 (0.2) | 0.1 (0.3) | -0.5 (0.3) | 0.4 (0.2) | 0.6 (0.2) |
| gemma2-27B | 88.3 (6.9) | 88.5 (6.9) | 93.0 (4.5) | 93.1 (4.6) | -0.0 (0.4) | -0.8 (0.4) | 0.6 (0.2) | 0.8 (0.3) | -0.1 (0.4) | -0.8 (0.3) | 0.6 (0.2) | 0.8 (0.3) |
| qwen2.5-32B | 85.5 (8.7) | 85.4 (9.2) | 89.1 (9.0) | 88.9 (9.7) | 0.2 (0.3) | -0.6 (0.3) | 0.4 (0.2) | 0.6 (0.2) | 0.2 (0.3) | -0.6 (0.3) | 0.4 (0.2) | 0.6 (0.2) |
| **Average** | 87.2 (7.4) | 87.0 (7.6) | 90.7 (7.1) | 90.6 (7.3) | 0.2 (0.4) | -0.7 (0.4) | 0.6 (0.2) | 0.8 (0.3) | 0.1 (0.4) | -0.7 (0.4) | 0.5 (0.2) | 0.8 (0.3) |

The results of APORIA indicate remarkable consistency between full and stemmed representations. In the original embedding space, separability ratios remain nearly identical, with mean values of 1.13 for full responses and 1.11 for stemmed responses, differing by less than 2%. To directly assess semantic preservation, we computed the cosine similarity between plain and stemmed response embeddings, obtaining a mean of 93% (std 5%) across all responses, confirming that stemming retains the vast majority of the information our framework operates on.

Similarly, Fisher-space separability shows minimal variation, with mean ratios of 7.26 for full responses and 7.53 for stemmed responses, representing a negligible difference of 3.7%. Model-specific patterns are also preserved: yi1.5-9B maintains the strongest separability in both conditions (9.53 full, 9.88 stemmed), while solar1-11B and phi4-14B remain among the weakest performers (5.29 and 5.82 full, 5.26 and 6.03 stemmed).

**Label Propagation of the Stemmed Responses** Table 9 extends the label propagation analysis (cf. Table 2) by decomposing APORIA-LP performance across full and stemmed responses while introducing a critical geometric diagnostic: the signed margin between genuine (G) and hallucinated (H) clusters in Fisher space. The signed margin quantifies the mean distance from each class to its own centroid relative to the opposing centroid, revealing the fundamental asymmetry in how the two response types organise in embedding space.

The consistency between full and stemmed responses confirms the robustness established in Table 8: accuracy differs by at most 0.3 percentage points across all models (e.g., yi1.5-9B: 92.5% full vs. 92.6% stemmed; deepseek-7B: 90.9% vs. 90.6%), with F1 scores showing comparable stability. This minimal variation indicates that APORIA-LP operates on semantic structure rather than lexical artefacts, supporting the generalisability of the Fisher-based approach across different

*Table 10.* Observed Wasserstein distances versus null hypothesis. Percentage of prompts exceeding null baseline, mean observed $\mathbb{E}[W]$ and null $\mathbb{E}[\bar{W}_{H_0}]$ distances (std in parentheses).

| Model | Full responses | | | Stemmed responses | | |
|---|---|---|---|---|---|---|
| | $\% W > \bar{W}_{H_0}$ | $\mathbb{E}[W]$ | $\mathbb{E}[\bar{W}_{H_0}]$ | $\% W > \bar{W}_{H_0}$ | $\mathbb{E}[W]$ | $\mathbb{E}[\bar{W}_{H_0}]$ |
| `mistral-7B` | 84.1 | 0.93 (0.47) | 0.36 (0.21) | 77.8 | 0.73 (0.38) | 0.33 (0.20) |
| `deepseek-7B` | 88.0 | 1.07 (0.71) | 0.37 (0.23) | 83.2 | 0.92 (0.66) | 0.36 (0.23) |
| `llama3-8B` | 86.2 | 0.94 (0.52) | 0.32 (0.22) | 84.1 | 0.80 (0.47) | 0.31 (0.22) |
| `gemma2-9B` | 81.5 | 0.78 (0.49) | 0.32 (0.20) | 77.2 | 0.62 (0.40) | 0.29 (0.19) |
| `yi1.5-9B` | 92.7 | **1.81 (0.83)** | **0.44 (0.23)** | **91.4** | **1.51 (0.76)** | **0.42 (0.23)** |
| `solar1-11B` | 83.7 | 0.93 (0.60) | 0.33 (0.21) | 78.8 | 0.73 (0.48) | 0.29 (0.19) |
| `phi4-14B` | 90.9 | 0.91 (0.58) | 0.28 (0.22) | 90.9 | 0.67 (0.44) | 0.24 (0.20) |
| `qwen2.5-14B` | 91.6 | 0.90 (0.48) | 0.31 (0.20) | 84.4 | 0.65 (0.38) | 0.29 (0.19) |
| `gemma2-27B` | 83.2 | 0.83 (0.58) | 0.34 (0.20) | 83.2 | 0.75 (0.57) | 0.32 (0.18) |
| `qwen2.5-32B` | **94.5** | 1.06 (0.57) | 0.34 (0.25) | 90.7 | 0.80 (0.46) | 0.30 (0.22) |
| **Average** | 87.6 | 1.02 (0.58) | 0.34 (0.22) | 84.2 | 0.82 (0.50) | 0.31 (0.20) |

preprocessing strategies.

The signed margin analysis reveals a clear geometric asymmetry that fundamentally characterises the embedding structure: across all models, genuine responses exhibit small margins near zero (average: 0.2 for full, 0.1 for stemmed), indicating that genuine clusters are compact and self-contained, with members residing close to their class centroid. In sharp contrast, hallucinated responses consistently display negative margins (average: $-0.7$ for both full and stemmed), with values ranging from $-0.4$ in `phi4-14B` to $-1.1$ in `deepseek-7B` and `yi1.5-9B`. These negative values indicate that hallucinated responses, as measured by the Wasserstein distance, are geometrically closer to the genuine cluster centroid than to their own hallucinated reference, despite belonging to the hallucinated class under ground-truth labelling.

### C.2. Further Results on the APORIA Structural Analysis

In this subsection we provide additional structural results for `APORIA` when applied to `SOCRATES-300K`.

**Extending the Structural Null Hypothesis Analysis** Table 10 reports the distributional differences between observed Wasserstein distances and the null hypothesis across all models. In particular, we show that Wasserstein distances between intra-genuine and intra-hallucinated distributions consistently exceed the null-hypothesis baseline obtained through random label permutation. The percentage of prompts showing separation above chance level ranges from 81.5% (`gemma2-9B`) to 94.5% (`qwen2.5-32B`), with an ensemble average of 87.6%, indicating that geometric differentiation is not an artefact of cherry-picked examples but rather a systematic property across diverse prompts and models.

The magnitude of separation is substantial. The mean observed Wasserstein distance across all models is $\mathbb{E}[W] = 1.02 \pm 0.58$, approximately three times larger than the null baseline $\mathbb{E}[\bar{W}_{H_0}] = 0.34 \pm 0.22$. `yi1.5-9B` exhibits the strongest distributional separation with $\mathbb{E}[W] = 1.81$, more than four times its null baseline of 0.44, while Gemma-9B shows the weakest separation at $\mathbb{E}[W] = 0.78$, still more than double its null value of 0.32. These results confirm that the dispersion asymmetry between genuine and hallucinated responses represents a robust geometric signature that significantly exceeds chance-level structure.

Figure 6 shows a representative example for `gemma2-9B`, in a very similar way to Figure 3, where prompts are ordered according to the observed Wasserstein distance between $\mathcal{D}_{GG}$ and $\mathcal{D}_{HH}$. The observed distance is compared with the distribution obtained under $H_0$, constructed by randomly permuting class labels, where prompts are ordered according to the observed Wasserstein distance between $\mathcal{D}_{GG}$ and $\mathcal{D}_{HH}$. For each prompt, the observed distance is compared with the distribution obtained under $H_0$, constructed by randomly permuting class labels. These results show graphically the observed Wasserstein distances reported in Table 10, where for `gemma2-9B` the 81% of the prompts exceed the null baseline.

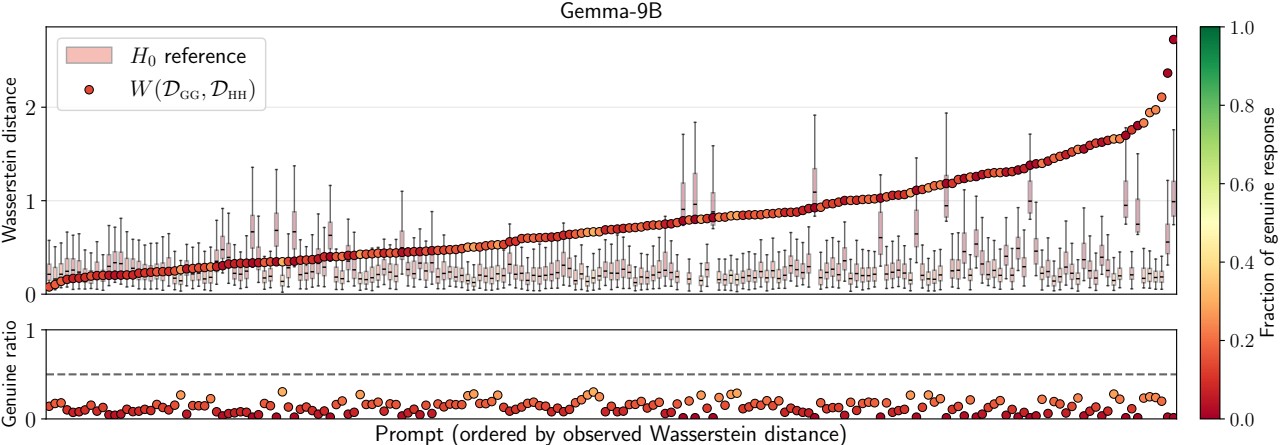

*Figure 6.* Statistical relevance of the distributional separation between $\mathcal{D}_{\mathrm{GG}}$ and $\mathcal{D}_{\mathrm{HH}}$ over the various prompts for `gemma2-9B`. (Top) The observed Wasserstein distance $W(\mathcal{D}_{\mathrm{GG}}, \mathcal{D}_{\mathrm{HH}})$, represented by ordered dots, is compared with the same distance evaluated on the null hypothesis $H_0$ obtained by permuting labels 100 times. (Bottom) Fraction of genuine responses is reported for each prompt, also encoded in the colour channel.

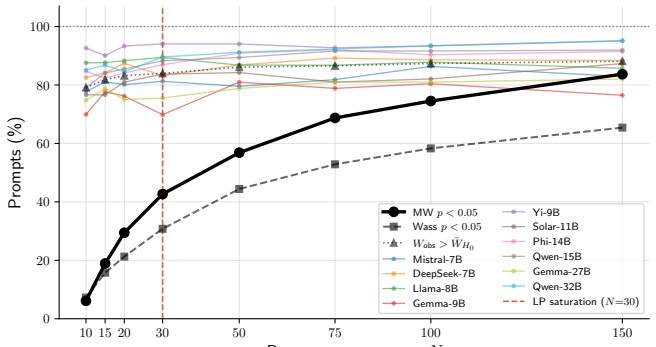

| $N$ | Wilcoxon (p sig) [%] | $W_{\mathrm{obs}}$ vs. $W_0$ (p sig) [%] | $W_{\mathrm{obs}} > W_0$ [%] |
|---|---|---|---|
| 10 | 6.1 | 7.3 | 79.1 |
| 15 | 18.9 | 15.8 | 81.8 |
| 20 | 29.4 | 21.3 | 83.2 |
| 30 | 42.7 | 30.8 | 84.0 |
| 50 | 56.8 | 44.4 | 86.1 |
| 75 | 68.7 | 52.8 | 86.6 |
| 100 | 74.5 | 58.3 | 87.3 |
| 150 | 83.7 | 65.4 | 88.1 |

*Figure 7.* Fraction of (model, prompt) pairs showing distributional separation between $\mathcal{D}_{\mathrm{GG}}$ and $\mathcal{D}_{\mathrm{HH}}$ as a function of subsample size $N$. For each pair and each $N$, three criteria are evaluated over 20 repeated subsamples: the Mann-Whitney $U$ test on $\mathcal{D}_{\mathrm{GG}}$ vs. $\mathcal{D}_{\mathrm{HH}}$ (solid, primary), the significance of Wasserstein distance against null permutations (dashed), and its point estimate $W_{\mathrm{obs}} > \bar{W}_{H_0}$ without a significance threshold (dotted). Numerical values are reported in the table on the right. The plot on the left also shows per-model Mann-Whitney rates (faint coloured lines) and the point $N = 30$ at which label-propagation performance saturates (cf. Figures 4 and 9).

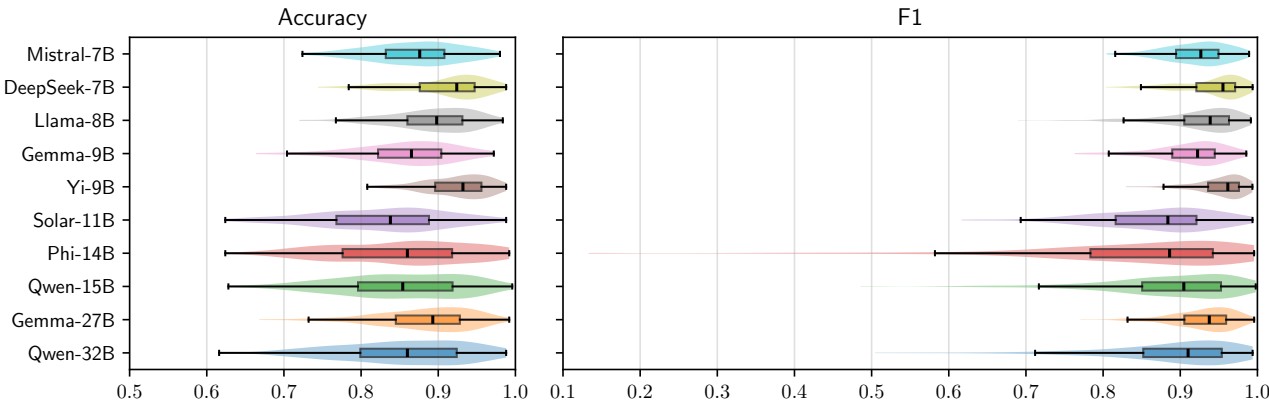

*Figure 8.* Label propagation performance across models and prompts. Box plots display the distribution of classification accuracy (left) and F1 score (right) across all 200 prompts for each model, evaluated using stratified shuffle split cross-validation to account for class imbalance at the prompt level. Each box represents the interquartile range of performance metrics computed independently for each prompt's response set.

**Sensitivity to the Response-Set Size** To contextualise the structural analysis, we study how Wasserstein separability is influenced by the size $N$ of the response set, comparing the observed Wasserstein distance $W(\mathcal{D}_{\mathrm{GG}}, \mathcal{D}_{\mathrm{HH}})$ with the null hypothesis for $N$ ranging from 10 to 150.

The results, in terms of Wasserstein-separability dominance and statistical significance, are displayed in Figure 7. At $N = 10$, 79.1% of prompts already exhibit clear separability, yet statistical significance is extremely low (7.3% at $p < 0.05$), reflecting the well-known challenge of distributional testing under small samples. At $N = 20$, the dominance rate rises to 83.2% and almost saturates at $N = 50$ (86.1%; cf. 88.1% at $N = 150$); however, statistical significance remains mild (21.3% and 44.4% respectively; cf. 65.4% at $N = 150$). Analogously, the significance of the distributional separation between $\mathcal{D}_{\mathrm{GG}}$ and $\mathcal{D}_{\mathrm{HH}}$ under the Mann–Whitney $U$ test (cf. Figure 2) remains limited at small $N$: 6.1%, 29.4%, and 56.8% at $N = 10, 20, 50$ respectively (cf. 83.7% at $N = 150$).

It is worth noting, however, that the structural analysis is intended as a characterisation tool rather than an online detection method. For the sensitivity of the label propagator to training-sample size, we refer to the next section.

### C.3. Further Results on the Label Propagation

In this subsection we provide additional results on `APORIA-LP` when applied to `SOCRATES-300K`.

**Further Details on the Label-Propagator Performance** Figure 8 reports the per-model breakdown behind Table 2, evaluated independently for each of the 200 prompts across all ten models. Given that many prompts exhibit substantial class imbalance—as visible in the distribution of hallucination rates across the dataset—F1-score provides a more robust assessment of classification quality than raw accuracy, since it accounts for both precision and recall without being inflated by majority-class prediction. The boxplots reveal consistently high performance across both metrics, with median accuracy values ranging from approximately 0.85 to 0.95 and F1-scores spanning a similar range. Notably, `mistral-7B` and `deepseek-7B` display both high median performance and low variability, while `phi4-14B` exhibits the widest spread in both metrics, consistent with the findings from the training-size analysis.

**Extending the Training-Size Sensitivity Analysis** Figure 9 presents the evolution of accuracy as a function of training-set size, visualising both the mean accuracy across prompts and its standard deviation. The experimental protocol mirrors that used for the F1 analysis: test sets remain fixed at one third of the available responses, while training size varies from 5 to 100 in steps of 5, with 10 random subsamplings when the training data does not exhaust the available pool. The trajectories reveal a consistent pattern across models: starting from the upper-left region (higher variance, lower mean accuracy), each model progresses toward the lower-right quadrant as more labelled examples become available, indicating simultaneous improvement in both central performance and stability. Most models achieve substantial accuracy gains and variance reduction within the first 30–50 labelled responses, after which improvements become marginal and the trajectories plateau. The clustering of final positions in the lower-right region demonstrates that the geometry-aware label-propagation

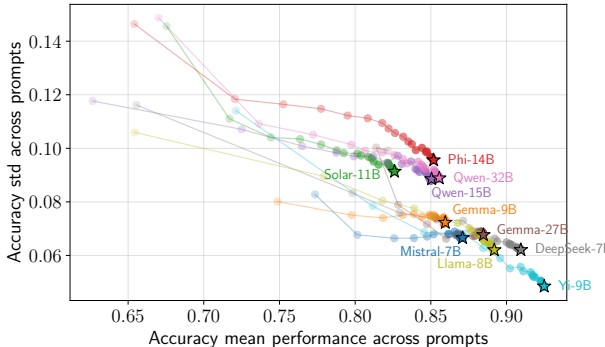

| N | F1 [%] | Acc [%] |
|---|--------|---------|
| 5 | 62.5 (19.0) | 66.6 (14.0) |
| 10 | 77.9 (14.9) | 74.6 (10.9) |
| 15 | 83.1 (12.4) | 78.6 (9.9) |
| 20 | 85.7 (11.2) | 81.1 (9.5) |
| 25 | 87.0 (10.8) | 82.6 (9.3) |
| 30 | 87.9 (10.6) | 83.7 (9.1) |
| 50 | 89.7 (9.5) | 85.9 (8.7) |
| 75 | 90.3 (9.2) | 86.8 (8.4) |
| 100 | 90.5 (8.7) | 87.0 (8.1) |

*Figure 9.* Evolution of the label-propagator accuracy with the training-set size, from 5 to 100 responses (step 5). The left plot shows model-wise results as mean ($x$-axis) versus standard deviation ($y$-axis): the further right and down, the better (cf. Figure 4). The right table reports aggregate values as mean (std).

*Table 11.* Hallucination rates across models on `CoQA-89K`.

| Model | Prompts | Total Responses | Hallucinations | Rate [%] |
|-------|---------|-----------------|----------------|----------|
| LLaMA-13B | 199 | 29,850 | 17,927 | 60.06 |
| LLaMA-7B | 199 | 29,850 | 20,169 | 67.57 |
| OPT-6.7B | 199 | 29,850 | 21,236 | 71.14 |
| **Total** | **598** | **89,550** | **66,262** | **66.26** |

approach yields comparably reliable performance across diverse architectures, with only `phi4-14B` and `yi1.5-9B` showing slightly more scattered trajectories—suggesting greater sensitivity to training-set composition for these particular models.

## D. Reproducing Results on CoQA

**CoQA-89K Generation** To further validate the results obtained on `SOCRATES-300K`, we constructed `CoQA-89K`, an additional bridge dataset using prompts from CoQA (Reddy et al., 2019), a large-scale dataset of 127k conversational question-answer pairs. Following the experimental design of `INSIDE`, we sampled 2.5% of the CoQA development split (originally 7,983 QA pairs), yielding 199 prompts; to enable a direct comparison with Chen et al. (2024), we adopted the same three small-sized (6.7B–13B) LLMs, namely `facebook/opt-6.7b`, `meta-llama/Llama-2-7b-hf`, and `meta-llama/Llama-2-13b-hf`[5]. For each prompt, we generated 150 independent responses per model under the same sampling configuration as `INSIDE`: nucleus sampling with top-$p = 0.99$, top-$k = 5$, and temperature $T = 0.5$. Hallucination labels were assigned using the `INSIDE` convention—ROUGE-L (Lin, 2004) $F$-measure against ground-truth answers, with a threshold of $0.5$ to classify each response as correct or hallucinated—applied here at the per-response level rather than to a single greedy decoding as in the original work. This reference-based labelling strategy provides an annotation-cost-free alternative to LLM-as-a-judge evaluation, requiring no external API calls.

The resulting hallucination rates per model are reported in Table 11 (cf. Table 7). Compared to `SOCRATES-300K`, the rates are notably lower, which can be attributed to two concurrent factors: (i) the different labelling methodology (ROUGE-L versus LLM-as-a-judge) and (ii) the nature of the prompts themselves. The latter observation suggests that time-sensitive factual queries may be inherently more prone to hallucination than conversational questions, consistent with the known difficulty of temporal reasoning in small-sized LLMs.

Following the same experimental protocol as before, we replicate both the structural analysis and the label propagation pipeline on `CoQA-89K`. The distributional separation results mirroring Figure 7 are shown in Figure 10, while the label

---

[5]While `opt-6.7b` is the very model used in Chen et al. (2024), the `meta-llama` variants are the closest available alternative to `INSIDE`'s original `decapoda-research` weights, which are no longer available on HuggingFace.

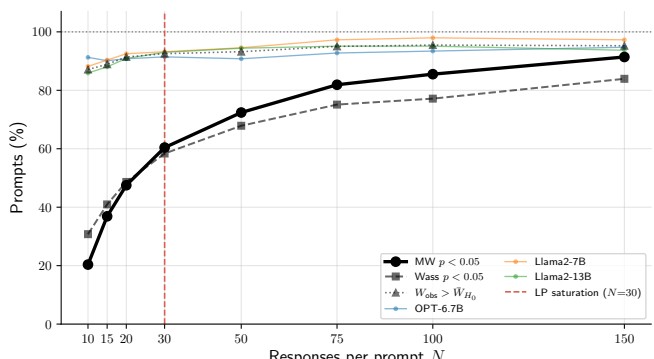

| $N$ | Wilcoxon (p sig) [%] | $W_{\text{obs}}$ vs. $W_0$ (p sig) [%] | $W_{\text{obs}} > W_0$ [%] |
|---|---|---|---|
| 10 | 30.8 | 20.4 | 87.1 |
| 15 | 41.0 | 36.9 | 88.9 |
| 20 | 48.6 | 47.5 | 91.4 |
| 30 | 58.4 | 60.4 | 92.5 |
| 50 | 67.9 | 72.4 | 93.2 |
| 75 | 75.1 | 81.9 | 95.0 |
| 100 | 77.1 | 85.5 | 95.5 |
| 150 | 83.9 | 91.4 | 95.2 |

*Figure 10.* Fraction of (model, prompt) pairs showing distributional separation between $\mathcal{D}_{\text{GG}}$ and $\mathcal{D}_{\text{HH}}$ as a function of subsample size $N$ when the methodology is evaluated on the `CoQA-89K` bridge dataset (plots mirror the structure of Figure 7).

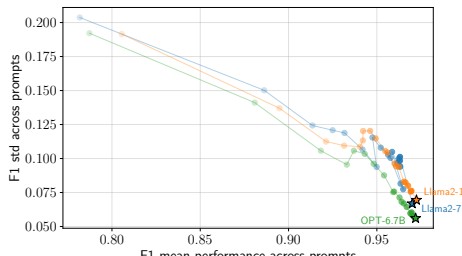

| $N$ | F1 [%] | Acc [%] |
|---|---|---|
| 5 | 78.7 (20.2) | 82.7 (14.3) |
| 10 | 88.6 (14.2) | 91.0 (8.8) |
| 15 | 91.5 (11.9) | 93.4 (6.8) |
| 20 | 92.8 (11.4) | 94.7 (5.7) |
| 25 | 93.4 (11.3) | 95.5 (5.0) |
| 30 | 94.0 (11.1) | 96.1 (4.5) |
| 50 | 95.7 (9.2) | 97.3 (3.5) |
| 75 | 96.4 (8.5) | 97.8 (3.0) |
| 100 | 97.1 (6.4) | 98.1 (2.7) |

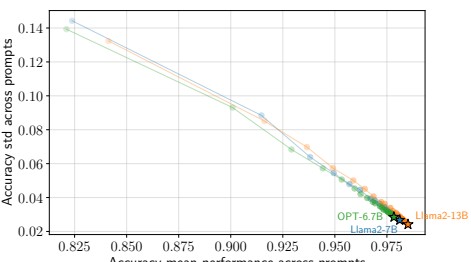

*Figure 11.* Label-propagator statistics when applied to the `CoQA-89K` bridge dataset (cf. Figure 9). Results are reported across training-set sizes from 5 to 100 responses (step 5). Left and right plots show model-wise F1 and accuracy as mean ($x$-axis) versus standard deviation ($y$-axis): the further right and down, the better (cf. Figure 4). The central table reports aggregate values as mean (std).

propagation performances mirroring Figures 4 and 9 are reported in Figure 11.

**Applying `APORIA` Structural Analysis on `CoQA-89K`** Figure 10 examines how Wasserstein separability depends on the response set size $N$ for `CoQA-89K`. At $N = 10$, $87.1\%$ of prompts already exhibit separability above the null baseline, yet statistical significance remains low (Wilcoxon: $30.8\%$; $W_{\text{obs}}$ vs. $W_0$: $20.4\%$), reflecting the well-known difficulty of distributional testing under small sample sizes. As $N$ grows, both significance measures increase steadily: at $N = 20$ the Wilcoxon significance reaches $48.6\%$ and $W_{\text{obs}} > W_0$ significance reaches $47.5\%$, while the fraction of prompts exceeding the null baseline rises to $91.4\%$. Near-saturation is observed around $N = 50$ ($67.9\%$, $72.4\%$, $93.2\%$), with further modest gains up to $N = 150$ ($83.9\%$, $91.4\%$, $95.2\%$). Compared to `SOCRATES-300K`, the significance levels at small $N$ are higher, consistent with the lower hallucination rates and more balanced class distributions observed in `CoQA-89K`.

As noted in Appendix C.2, the structural analysis is intended as a characterisation tool rather than an online detection method; the sensitivity of the label propagator to training sample size is discussed in the following section.

**Applying `APORIA-LP` on `CoQA-89K`** Label-propagation performances are reported in Figure 11. Notably, when compared with the results on `SOCRATES-300K`, `APORIA-LP` attains even stronger scores; this can be attributed to two concurrent factors: the nature of CoQA prompts, whose shorter and more direct answers tend to cluster more tightly in embedding space, and the ROUGE-L $\geq 0.5$ labelling criterion inherited from `INSIDE`. Remarkably, the label propagator reaches usable accuracy ($> 90\%$) from as few as $N = 10$ labelled samples, compared to $N \sim 75$ required on `SOCRATES-300K`, further confirming the efficiency of the proposed framework on datasets structurally different from our main testbed.

**Cross-Dataset Results on `CoQA-89K`** Table 12 reports the results of `APORIA` and `APORIA-LP` on each of the three models used to generate the `CoQA-89K` bridge dataset, when compared with the original Chen et al. (2024)'s results and `INSIDE`'s results on `CoQA-89K`. Fisher separability reaches an average of $79.65\times$, approximately $10.97\times$ higher than the $7.26\times$ reported for `SOCRATES-300K` in Table 1, consistent with the tighter geometric structure observed on CoQA prompts. Regarding the AUC-ROC comparison, our re-execution yields results closely aligned with those reported

Table 12. Cross-dataset results on `CoQA-89K` for the three models evaluated in `INSIDE` (Chen et al., 2024).

| Model | APORIA structural | | APORIA-LP | | INSIDE | |
|---|---|---|---|---|---|---|
| | W ratio | Fisher sep. | Acc [%] | F1 [%] | AUC-ROC [%] | (Chen et al., 2024) [%] |
| LLaMA-7B | 2.17 | 66.92 | 98.1 | 96.6 | 78.2 | 80.8 |
| LLaMA-13B | 2.26 | 107.74 | 98.5 | 97.3 | 83.8 | 80.8 |
| OPT-6.7B | 1.90 | 64.30 | 97.8 | 97.3 | 69.0 | 77.5 |
| **Average** | **2.11** | **79.65** | **98.1** | **97.1** | **77.0** | **79.7** |

Table 13. Comparison of label-propagation methods across embedding spaces. Results are mean (std) over models and prompts.

| Method | Fisher 1D | | SBERT 384D | |
|---|---|---|---|---|
| | Acc [%] | F1 [%] | Acc [%] | F1 [%] |
| APORIA-LP (Wass.) | 87.0 (8.1) | 90.5 (8.8) | 72.7 (14.5) | 79.9 (14.3) |
| Centroid (Euclidean) | 86.0 (8.4) | 89.7 (9.0) | 80.1 (10.8) | 84.9 (11.1) |
| Centroid (Cosine) | – | – | 80.0 (10.7) | 84.8 (11.0) |
| Linear Regression | 88.3 (7.7) | 91.4 (9.2) | 87.8 (8.0) | 91.1 (9.1) |
| SVM | 88.0 (7.8) | 91.3 (8.9) | 83.0 (9.3) | 87.7 (9.3) |
| kNN-5 | 88.0 (7.8) | 91.3 (9.0) | 86.6 (8.7) | 89.6 (11.6) |

in `INSIDE` for `LLaMA-7B` and `LLaMA-13B`, with a slightly larger discrepancy for `OPT-6.7B`. These differences are expected given the substantial gap in experimental scale: our `CoQA-89K` subset comprises 199 prompts with $K = 150$ responses each, whereas `INSIDE` operates on 7,983 prompts with only 10 responses per prompt.

## E. Ablating the Wasserstein Score With Other Methodologies for Label Propagation

We performed a comparative ablation over `APORIA-LP` with different classification strategies, to assess the effectiveness of the proposed Wasserstein label-propagation scheme—particularly when paired with and without the 1D Fisher reduction. The objective is to determine whether alternative classical solutions might perform better.

Results are presented in Table 13, where `APORIA-LP` is compared with ten label-propagation strategies on `SOCRATES-300K`. On the Fisher projection ($1D$), all classifiers—Centroid Euclidean, Linear Regression, Support Vector Machines (SVM), and 5-Nearest Neighbours (kNN-5)—cluster tightly around 86–88% accuracy and 89%–91% F1. This confirms that the Fisher direction captures the discriminative signal so cleanly that the choice of classifier is nearly interchangeable; in this sense, the structural finding provided by `APORIA` is the real contribution, and the `APORIA-LP` method is secondary. In the full SBERT ($384D$) space, by contrast, performance is more variable: the Linear Regression remains strong (87.8%/91.1%) while SVM and centroid-based methods (here also applied in cosine-variant) degrade, confirming the value of the Fisher projection in isolating the discriminative signal from noise dimensions. Finally, we also tested Linear Regression on the three-dimensional feature space derived from centroid distances, which performs well ($87.4 \pm 8.1\%/90.7 \pm 10.9\%$, not shown in table), but still comparable with the other solutions on the Fisher projection.

Crucially, `APORIA` links the classification decisions to the intra-class geometry ($D_{GG}, D_{HH}$), providing an interpretation not present in any standard classifier reported in the literature. It is also worth mentioning that `APORIA` remains computationally competitive (less than $2\times$ the runtime of the other methodologies).

## F. Comparison of Projection Geometries for Label Propagation

We performed a comparative evaluation of different projection strategies in conjunction with the proposed Wasserstein label propagation scheme. The aim was to determine whether more complex or higher-dimensional embeddings provide measurable benefits over a simple supervised linear projection.

Table 14 summarises the results for each method across multiple choices of embedding dimension, with all experiments using the full training set and averaged over fixed stratified test splits.

*Table 14.* Performances of different projection methods (Fisher vs. UMAP vs. wPCA vs. EP) over the label propagation scheme.

| Method | # Components | Accuracy [%] | F1 [%] | $M$ | $|M|$ | % Agree |
|---|---|---|---|---|---|---|
| Fisher | 1 | **86.9** (8.1) | **90.5** (8.8) | -0.60 (0.55) | 0.78 (0.29) | 100.0 (0.0) |
| UMAP | 1 | 67.9 (16.8) | 70.1 (23.6) | -0.05 (11.86) | 13.11 (4.45) | 71.5 (16.8) |
| | 2 | 73.7 (16.3) | 75.2 (23.4) | -0.73 (9.86) | 11.63 (2.99) | 77.7 (15.9) |
| | 3 | 73.9 (16.0) | 75.4 (23.3) | -0.86 (9.80) | 11.61 (2.85) | 77.9 (15.6) |
| | 5 | 73.8 (16.0) | 75.2 (23.4) | -0.87 (9.85) | 11.66 (2.81) | 77.8 (15.6) |
| | 10 | 73.9 (15.9) | 75.2 (23.4) | -0.94 (9.80) | 11.56 (2.82) | 77.9 (15.4) |
| | 15 | 73.7 (15.8) | 75.1 (23.2) | -0.95 (9.83) | 11.58 (2.86) | 77.7 (15.4) |
| wPCA | 1 | 69.6 (16.9) | 75.9 (17.3) | -0.03 (0.05) | 0.05 (0.04) | 71.8 (17.7) |
| | 2 | 74.1 (14.8) | 80.6 (14.7) | -0.04 (0.05) | 0.05 (0.04) | 75.5 (15.7) |
| | 3 | 74.0 (14.5) | 80.6 (14.5) | -0.04 (0.05) | 0.05 (0.04) | 74.6 (15.3) |
| | 5 | 70.8 (15.0) | 77.9 (15.2) | -0.04 (0.05) | 0.05 (0.04) | 70.7 (15.7) |
| | 10 | 63.2 (16.8) | 70.1 (17.8) | -0.03 (0.05) | 0.05 (0.03) | 63.2 (17.6) |
| | 15 | 58.2 (17.9) | 64.5 (19.3) | -0.02 (0.05) | 0.05 (0.03) | 58.6 (18.8) |
| EP | 1 | 61.8 (14.3) | 69.8 (15.5) | -0.42 (0.94) | 0.80 (0.70) | 62.6 (14.4) |
| | 2 | 64.6 (12.9) | 72.9 (14.0) | -0.42 (0.78) | 0.70 (0.58) | 65.3 (13.1) |
| | 3 | 66.9 (12.6) | 75.0 (13.5) | -0.48 (0.78) | 0.73 (0.60) | 67.7 (13.0) |
| | 5 | 67.9 (12.6) | 76.1 (13.0) | -0.48 (0.72) | 0.70 (0.56) | 68.3 (12.8) |
| | 10 | 70.1 (13.3) | 77.9 (13.6) | -0.49 (0.62) | 0.66 (0.50) | 70.4 (13.6) |
| | 15 | 71.2 (13.5) | 78.8 (13.7) | -0.51 (0.60) | 0.66 (0.48) | 71.3 (14.0) |

Accuracy and F1 are also reported visually in Figure 12 as a function of the number of extracted components.

**Linear Fisher Projection** The Fisher projection consistently achieves the highest accuracy and F1 score while operating in a single dimension (86.9% accuracy, 90.5% F1), and exhibits perfect agreement with itself as expected. This indicates that the bulk of the discriminative information between genuine and hallucinated responses can be captured by a single linear direction aligned with the difference of class means. The low variance across splits further suggests that this projection is both robust and stable.

**Nonlinear and Unsupervised Projections** UMAP attains moderate accuracy and F1 scores (approximately 74% and 75%, respectively) with some improvement at three dimensions (after which performance remains constant). However, Wasserstein margins are extremely large and highly variable (particularly compared to the signed ones), possibly reflecting strong geometric distortions introduced by the nonlinear embedding. These results suggest that while UMAP can recover the general separation, its calibration and interpretability are reduced compared to the Fisher projection.

Whitened PCA (wPCA) and Random Projections (EP) generally perform worse, even as the number of components increases. This confirms that preserving variance or random projections of the space is insufficient when the relevant signal is primarily discriminative rather than generative. Interestingly, wPCA achieves a modest peak at 2–3 components (accuracy $\sim 74\%$, F1 $\sim 81\%$), before declining at higher dimensions.

**Agreement Between Methods** To further characterise the decision boundaries, we computed the fraction of test points for which each method agrees with the Fisher projection. UMAP shows relatively high agreement (approximately 78%), indicating that nonlinear methods largely recover the same separation identified by the Fisher direction. In contrast, wPCA and EP exhibit lower agreement (around 70–71%), consistent with their lower predictive performance.

Overall, these results reinforce that a Fisher-based linear geometry provides an effective and robust representation for Wasserstein label propagation in this setting. It outperforms more complex projections, remains low-dimensional, and yields interpretable margins and consistent decisions, making it a natural baseline for future experiments.

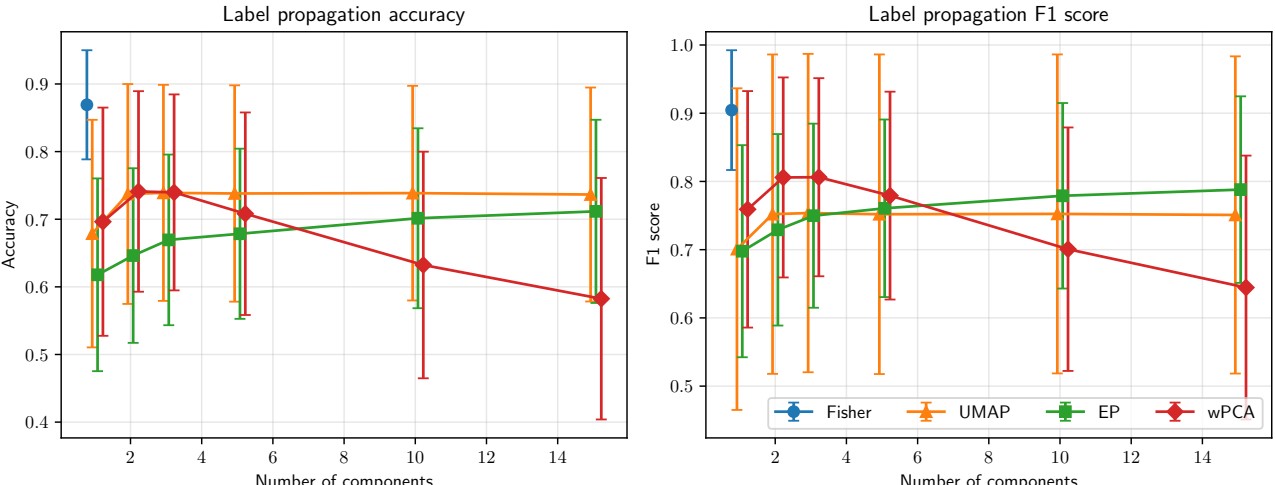

*Figure 12.* Label propagation accuracy and F1 score vs. the number of components for different projection methods.

