# OpenReview forum: "A Geometric Analysis of Small-sized Language Model Hallucinations"
_ICML.cc/2026/Conference — ICML 2026 regular_

### Official Review · Reviewer_4mi3 · 2026-03-12

**Soundness:** 3
**Presentation:** 2
**Significance:** 2
**Originality:** 2
**Overall Recommendation:** 2
**Confidence:** 4

**Summary:**

The paper builds on the observation that genuine (correct) responses tend to cluster more tightly in embedding space, whereas hallucinated responses exhibit weaker semantic cohesion. To support this observation, the authors conduct an extensive set of experiments on smaller-scale models. They also release the dataset generated for this study, which is a valuable contribution that may facilitate further research and reproducibility. In addition, the paper proposes a detection algorithm based on this observation and evaluates its effectiveness on the released dataset.

**Compliance With Llm Reviewing Policy:**

Affirmed.

**Final Justification:**

My main concerns on novelty and lack of comparison with baselines on public datasets are not resolved. Thus, I maintain my original score.

**Key Questions For Authors:**

see weaknesses

**Limitations:**

yes

**Strengths And Weaknesses:**

Strength:
The paper conducts an extensive set of benchmarks to support and validate its findings, which helps provide empirical evidence for the proposed claims. The experimental evaluation spans multiple settings, contributing to a more comprehensive assessment of the approach. In addition, the authors publicly release the dataset used in their study. This is a valuable contribution, as it enables reproducibility and may serve as a useful resource for future research in this area.

Weakness:
1. One major concern is the level of novelty. The key observation—that correct answers cluster in embedding space—has already been demonstrated in well-established prior work [1]. Although [1] supports this observation using a different clustering metric, the central conclusion remains essentially the same. As a result, it would be helpful for the authors to more clearly articulate how their contribution differs from or advances beyond this existing finding.

2. The evaluation of the proposed metric appears to lack sufficient scientific rigor. The study does not include comparisons with established baseline methods, despite the extensive literature available in this area. Additionally, the evaluation is limited to a dataset generated by the authors, with no experiments on independent datasets. Including baseline comparisons and testing on external datasets would strengthen the validity and generalizability of the results.


[1] Chen, Chao, Kai Liu, Ze Chen, Yi Gu, Yue Wu, Mingyuan Tao, Zhihang Fu, and Jieping Ye. "INSIDE: LLMs' Internal States Retain the Power of Hallucination Detection." In The Twelfth International Conference on Learning Representations.

---

> ### Author Rebuttal · Authors · 2026-03-31
>
> We thank the Reviewer for their feedback and for pointing us to INSIDE (Chen et al., ICLR 2024).
> We respectfully argue that the novelty concern rests on a conflation of two distinct contributions, and that the claim of insufficient rigour is not supported by evidence.
>
> ## Q1. Novelty
> The Reviewer states that our key observation (correct answers clustering in embedding space) was already demonstrated in INSIDE.
> That said, we believe this conflates two findings that share a high-level intuition but differ fundamentally.
>
> INSIDE's evaluation assigns a single correctness label per question, not per response.
> Correctness is determined comparing a single primary answer (ie greedy decoding for temperature T=0) to the ground truth (GT) via ROUGE-L≥.5, see App.H (where LLMAns is evaluated against GT independently of BatchGenerations) and confirmed by inspecting the GitHub repo.
> The K=10 sampled responses are exclusively used for evaluating the detection score (EigenScore) to measure the semantic dispersion of the generation and never individually assessed, as such:
>  1. Overconfident hallucinations are not captured (no dispersion in the responses, see Section 3.2 of INSIDE);
>  2. INSIDE cannot distinguish between fully-hallucinated prompts and ones where some responses are actually genuine.
>
> Most importantly, it causes the model to flag heavily hallucinated questions as hallucinated prompts, despite having the knowledge to answer them correctly (cf. INSIDE App.H1, Q5).
>
> Conversely, our work focuses precisely on within-prompt heterogeneity:
> We show average H/G disagreement of ~20% (std 20%) on our "bridge" dataset (see below) and reveale that genuine and hallucinated responses have distinct geometric signatures even when they coexist for the same prompt;
> This regime is invisible to INSIDE's per-question evaluation.
>
> Further distinctions follows:
> *(i)* INSIDE requires *white-box* access to LLM internal states while we work black-box and cross-model (cf. Rev neJx, Q4).
> *(ii)* INSIDE proposes an *aggregate* divergence measure compressing the batch into a single scalar. Our geometric analysis separately computes D_GG, D_HH, and D_GH, revealing structural asymmetries (Table 3) that no aggregate score can capture.
> *(iii)* INSIDE performs unsupervised detection while we do supervised Label Propagation (LP).
> *(iv)* INSIDE operates on 5 models without releasing a public full dataset, while we release a public dataset of 300K tagged response generated across 10 models.
>
> To sum up, we acknowledge the shared intuition that semantic consistency relates to correctness.
> Our main contribution, however, lies in the geometric characterisation of *how* responses cluster, *how* G and H differ structurally within the same prompt, and *how* this enables LP, questions that INSIDE's per-question framework does not and cannot address.
>
> ## Q2. Evaluation rigour
> We respectfully disagree that our evaluation "lacks sufficient scientific rigour."
> Our study spans 300K responses (1K manually validated) on 10 models/200 prompts whith multiple ablation studies and a public code-base & dataset release.
> That said, we agree that the submission would benefit from baseline comparisons and cross-dataset evaluation, below addressed.
>
> *Baselines:*
> See our response to Rev 82cu (Q2) for the full table including standard classifiers and literature methods, along with EigenScore applied to our dataset.
> In particular, regarding EigenScore, we acknowledge that:
>  - we used the external SBERT embeddings since we did not capture the internal tokens (white-box access) during dataset generation.
>  - we defined prompt-level correctness via majority vote (H/G if ≥50% of responses are H/G) since we do not have a GT.
> 5 models exhibit near-total hallucination rates across all prompts, rendering AUROC undefined, offering a regime substantially more challenging w.r.t. INSIDE's benchmarks.
> For the remaining 5, AUROC ranges from 11.1% to 97.7%, averaging at 62.2% whith Acc/F1 averaging at 66.0%/69.2% (under G-Mean, cf.INSIDE App.H), well below expectations (cf. 90.5% F1 our LP on the same data, Rev 82cu).
>
> *Cross-dataset replication.*
> We have generated a "bridge" dataset of N=150 responses for 199 CoQA prompts (`--fraction-of-data-to-use .025`) using INSIDE's three evaluation models (replacing LLaMA-xB with LLaMA-2-xB as the original weights are unavailable), following their sampling parameters.
> Per-response correctness labels were obtained via ROUGE-L ≥ 0.5 against CoQA GT (cf. INSIDE), now applied to each individual response.
> Results of our LP on SBERT and EigenScore AUCROC on internal tokens (cf. INSIDE Sec. 3.1) follows.
>
> |Model|W ratio|Fisher sep.|LP acc|LP F1|EigenScore AUCROC|EigenScore ref|
> |-|-|-|-|-|-|-|
> |LLaMA-7B|2.17|66.92|98.1|96.6|78.2|80.8|
> |LLaMA-13B|2.26|107.74|98.5|97.3|83.8|80.8|
> |OPT-6.7B|1.90|64.30|97.8|97.3|69.0|77.5|
>
> AUCROC values from INSIDE paper are reported as a reference, offering small variations as prompts are 199 w.r.t. 7,983 and K=150 w.r.t. K=10.

---

> > ### Author Rebuttal · Reviewer_4mi3 · 2026-04-02
> >
> > Thank you for the rebuttal.
> >
> > I remain concerned that the design differences between INSIDE and the proposed method are not sufficient to justify a claim of novelty. The differences highlighted in the rebuttal, e.g. use of majority vote, number of samples, does not reflect fundamental design difference in my opinion.
> >
> > Thank you for providing a comparison with INSIDE on CoQA.  But echoing with baseline requirement in Q2 of reviewer 82cu, there is extensive prior work in this area.  All of the mentioned methods are evaluated on a diverse set of public datasets and compared against prior work. While conducting experiments at this scale within the short span of the rebuttal is not feasible, I believe that a larger-scale empirical evaluation is necessary to understand the practical value of the algorithm.

---

> > > ### Author Response · Authors · 2026-04-05
> > >
> > > We thank the Reviewer for the continued engagement.
> > > However, we respectfully note that the rebuttal response mischaracterises our novelty argument, and we would like to correct this before the discussion concludes.
> > >
> > > ## On the claimed "design differences"
> > > The Reviewer states that
> > > >"the differences highlighted in the rebuttal, e.g. use of majority vote, number of samples, does not reflect fundamental design difference."
> > >
> > > We must clarify: majority vote and number of samples were never presented as novelty claims.
> > > They are implementation details of how we adapted EigenScore to our dataset (majority vote for prompt-level ground truth; N=150 as sampling budget).
> > > The actual novelty arguments, presented as the central points of Q1, are:
> > >
> > > ### 1. White-box vs. black-box
> > > INSIDE requires access to internal hidden states; their own paper acknowledges this as a limitation.
> > > Our method operates on external sentence embeddings, enabling analysis of **API-only and closed-source models**, and cross-model comparisons in a shared embedding space, enabling future research directions (cf. eg. Rev neJx Q2).
> > > This is not a parameter choice; it is an architectural constraint that determines what analyses are possible.
> > >
> > > ### 2. Per-response vs. per-question analysis
> > > INSIDE assigns one label per question (from the greedy answer) and computes one scalar per question (EigenScore).
> > > It never examines how individual responses within a batch relate to each other.
> > > Our work labels every response individually and studies the *geometric structure* of the response distribution: D_GG, D_HH, D_GH, Fisher separability, signed margins.
> > > This is not a difference in "number of samples"; it is a fundamentally different scope and analytical object.
> > >
> > > ### 3. Class-conditional geometry vs. aggregate divergence
> > > INSIDE's EigenScore measures total semantic dispersion.
> > > It cannot distinguish a prompt where 80% of responses are correct from one where 80% are hallucinated, as both can produce identical EigenScores if overall dispersion is similar.
> > > Our analysis separately characterises the geometry of each class and reveals structural asymmetries (e.g., hallucinated responses being closer to the genuine centroid than to their own, Table 3) that no aggregate measure can capture.
> > > This finding (the asymmetric margin structure) has no precedent in INSIDE or, to the best of our knowledge, in any prior work.
> > >
> > > ## On the key observation
> > > The Reviewer states that INSIDE "demonstrated" that "correct answers cluster in embedding space."
> > > INSIDE showed that *low EigenScore correlates with correctness*, i.e., when all responses are semantically similar, the greedy answer tends to be right (yet not always, as shown by overconfident hallucinations).
> > > This is a correlation between aggregate dispersion and a per-question binary label: neither a proof, nor a structural characterisation.
> > > Our work shows something qualitatively different: that *within* a set of responses to the same prompt, genuine and hallucinated responses form geometrically distinct clusters with measurable structural properties and (departing from the key observation) usable separability.
> > > INSIDE never separates responses into classes, never computes intra-class distances, and never examines the geometry of the separation.
> > > The claim that this was "already demonstrated" conflates a detection heuristic with a structural characterisation.
> > >
> > > Furthermore, INSIDE's per-question design does not acknowledge the coexistence of hallucinated and genuine responses within the same prompt, which is the central phenomenon our work studies. This distinguishes our perspective from the implicit assumption (common in detection-oriented work) that hallucination reflects missing knowledge, rather than retrieval instability (cf. our Discussion, Insight 1).
> > >
> > > ## On the scale of evaluation
> > > We fully agree that broader evaluation strengthens any contribution.
> > > However, our study already spans 10 models, 200 prompts, 300K responses, with manual validation, multiple ablations, and a public dataset; this is a scale that compares favourably with INSIDE (5 models, 4 datasets [which serve as ground-truth sources rather than evaluation benchmarks], no public dataset) and SelfCheckGPT (hallucination evaluated on GPT-3+WikiBio only, with LLaMa and gpt-3.5 for auxiliary roles and 9 proxy LLMs).
> > > This scale is further strengthened by the rebuttal, providing cross-dataset replication on CoQA (>97% F1), 10+ baseline comparisons, and structural ablation down to N=10.
> > > This constitutes a rigorous foundation for a contribution whose primary claim is structural/analytical rather than a detection algorithm competing on leaderboards.
> > >
> > > We hope that this final clarification highlights that the novelty of our work lies in the analytical framework rather than in the implementation parameters, ie. a per-response, class-conditional, black-box geometric characterisation of hallucination that is fundamentally distinct from INSIDE's per-question aggregate detection approach.

---

### Official Review · Reviewer_82cu · 2026-03-12

**Soundness:** 2
**Presentation:** 3
**Significance:** 3
**Originality:** 4
**Overall Recommendation:** 4
**Confidence:** 3

**Summary:**

This paper investigates the geometric structure of the small model illusion. Studies have shown that for multiple texts generated from the same prompt, the factual correct texts form closer clusters than the hallucinatory texts, and when projected onto the Fisher discriminant direction, they exhibit strong one-dimensional separability. Based on this assumption, the paper obtained a total of 300,000 responses by running 10 small models, 200 prompts, and 150 sampling times for each prompt.

**Compliance With Llm Reviewing Policy:**

Affirmed.

**Key Questions For Authors:**

1.The abstract states that "we proved" the clustering/separability hypothesis. Which theorem is it referring to? What are the assumptions (such as the distribution form, covariance structure) and where is the proof? If no proof has been provided, please revise this claim or provide a formal derivation.

2.Could you report the comparison with (a) the simple geometric baseline (centroid classifier, 384-dimensional SVM/LR, cosine to centroid + length features) and (b) the standard detectors (semantic entropy, SelfCheckGPT, Shapley uncertainty), using your dataset?

3.How does this method perform when generating significantly fewer samples each time (for example, 10-20)? Does this approach come closer to the actual operational budget? Please include a sample size ablation experiment. I believe this is crucial for the final judgment.

**Limitations:**

yes

**Strengths And Weaknesses:**

**Strengths**

1.The problem setting is interesting. The paper does not merely discuss hallucination in a general way, but focuses on the response structure under the condition of "multiple samplings of the same prompt". This is closer to the random nature of the generative model than the traditional single-response discrimination and thus has greater research value.

2.The experimental scale is relatively solid in this task. There are 10 public small models, 200 prompts, and 150 samples for each prompt, resulting in a total of 300,000 responses. The data volume is sufficient to support the structural statistics at the prompt level.

3.The expression is clear. The overall process, structural analysis, and the classification method based on projection have all been clearly introduced.

**Weaknesses**

1.The abstract and the statement claim that "we have proved" the core hypothesis, but no formal theoretical proof or the underlying assumptions for its validity have been provided.

2.There is a lack of a sufficiently strong baseline comparison between "geometric discovery" and "classifier effectiveness". The author demonstrated that the FDA + Wasserstein propagation method works, but did not adequately explain whether it is truly superior to the more direct and natural baseline. For instance, on the same SBERT embedding, what would be the results if we used logistic regression, linear SVM, kNN, naive nearest centroid, Gaussian classifier?

---

> ### Author Rebuttal · Authors · 2026-03-31
>
> We thank the Reviewer for the clear and focused feedback.
> We are pleased the Reviewer found the problem setting interesting and the experimental scale solid.
> We address each question below.
>
> ## Q1. Use of "proved"
> We thank the Reviewer for pointing out the overclaim the term "proved" carries.
> The term was used in its broader sense of "showing through extensive experiments" rather than its formal mathematical sense.
> No formal theorem or proof is claimed.
> We will replace "proved" with "empirically showed" in the abstract of the camera-ready version.
>
> ## Q2. Baseline comparisons
> We appreciate this suggestion and have now run the requested baseline comparisons, both on Fisher space (1D) and SBERT embedding (384D).
> Regarding other baselines we run:
>  - SelfCheckGPT: for each (model, prompt) we randomly picked 20 target responses and, for each of them, we draw 10 stochastic evidence passages from the remaining unpicked, computing per-model aggregate scores.
>  - Semantic Entropy: we adapted the prompt-wise methodology to a per-response detection; for each (model, prompt), we picked 20 responses, group them into semantic equivalence classes via bidirectional NLI with DeBERTa-v2-xlarge-mnli, as per the original work.
>  - EigenScore: see Rev 4mi3 Q2.
>
> |Method|Space|Acc|F1|
> |-|-|-|-|
> |Wass. (ours)|Fisher 1D|87.0 (8.1)|90.5 (8.8)|
> |||||
> |Centr. Eucl.|Fisher 1D|86.0 (8.4)|89.7 (9.0)|
> |LR|Fisher 1D|88.3 (7.7)|91.4 (9.2)|
> |SVM|Fisher 1D|88.0 (7.8)|91.3 (8.9)|
> |kNN-5|Fisher 1D|88.0 (7.8)|91.3 (9.0)|
> |||||
> |Wass.|SBERT 384D|72.7 (14.5)|79.9 (14.3)|
> |Centr. Eucl.|SBERT 384D|80.1 (10.8)|84.9 (11.1)|
> |Centr. Cos.|SBERT 384D|80.0 (10.7)|84.8 (11.0)|
> |LR|SBERT 384D|87.8 (8.0)|91.1 (9.1)|
> |SVM|SBERT 384D|83.0 (9.3)|87.7 (9.3)|
> |kNN-5|SBERT 384D|86.6 (8.7)|89.6 (11.6)|
> |LR|Centroid-feat 3D|87.4 (8.1)|90.7 (10.9)|
> |||||
> |EigenScore (§)|SBERT 384D|66.0 (30.7)|69.2 (29.8)|
> |SelfCheckGPT-NLI (§)|-|83.1 (5.5)|90.5 (3.3)|
> |Semantic Entropy (§)|NLI clusters|74.7 (5.4)|83.2 (5.2)|
>
> *(§) results extracted on reduced-size dataset (10% of prompts, ~30k responses) due to computational time. Full results will be available during interactive discussion.*
>
> On the Fisher projection, all classifiers cluster tightly around 86–88% accuracy and 89% F1, with predictions agreeing on over 95% of test instances.
> This confirms that the Fisher direction captures the discriminative signal so cleanly that the choice of classifier is nearly interchangeable; in this sense, the structural finding is the real contribution, and the Label Propagator (LP) method is secondary.
> In full SBERT space, performance is more variable: LR remains strong (87.8%/91.1%) while SVM and centroid-based methods degrade, confirming the value of the Fisher projection in isolating the discriminative signal from noise dimensions.
> However, we chose the Wasserstein formulation not for raw accuracy but for its direct connection to the structural analysis: it produces signed distributional margins (Table 3) that quantify how deeply a response sits within its class distribution, linking classification decisions to the intra-class geometry (D_GG, D_HH) that is the core of our work: no standard classifier naturally provides this interpretability.
> Furthermore, Wasserstein LP remains computationally competitive (less that 2x comput. time).
> The EigenScore baseline (66.0%/69.2%) performs substantially below all supervised methods, confirming that aggregate divergence is not a substitute for class-conditional analysis.
> SelfCheckGPT-NLI achieves remarkable performance for an unsupervised method, offering comparable F1, yet lower overall accuracy.
> Semantic Entropy enjoys good performance, further confirming that unsupervised consistency-based methods lack the per-response resolution that our class-conditional geometric analysis provides.
>
> ## Q3. Sample-size ablation
> We agree with the Reviewer that ablation studies are essential to assess the effectiveness of our work.
> Figures 4 and 8 already provide ablation for the LP task, showing that performance saturates around 30–50 labelled samples.
> To complement this, we now provide ablation for the *structural analysis* itself:
> in particular, we subsampled responses at various N per prompt and recomputed the Wasserstein distance W(D_GG, D_HH) relative to the null permutation.
> At N=10, 74.1% of prompts still show great separability, yet statistical significance is extremely low (7.7% p<0.05), reflecting the well-known challenge of distributional testing under small sets.
> At N=20, percentage rise to 81.0% and almost saturates at N=50 with 85.1% (cf. 88.3% at N=150), however significance remains overall mild at 21.9%, 43.8% (cf 65.9%).
> Analogously, distributional separation between D_GG and D_HH significance (Wilcoxon test, cf. Fig 2) remains limited: 6.1%/20.6%/44.6% at N=10/20/50 (cf. 83.7%, N=150).
> We note, however, that the structural analysis is intended as a characterisation tool; the Wass LP already demonstrates effectiveness from 30 labelled samples.

---

> > ### Author Rebuttal · Reviewer_82cu · 2026-04-03
> >
> > I think the author has partially solved my problem. However, I have kept my original score of "weakly accepted" unchanged.

---

> > > ### Author Response · Authors · 2026-04-07
> > >
> > > We thank the Reviewer for acknowledging that the concerns have been partially addressed and for maintaining the positive score.
> > > We are glad the baseline comparisons, sample-size ablation, and other new results contributed to this assessment.
> > >
> > > Since the Reviewer indicates partial resolution with follow-up questions, we would like to proactively address the points we believe may remain open before the discussion concludes:
> > >
> > > ## Q2 follow-up. Baselines
> > > We here report the full statistics for the incomplete baselines from the previous answer (Table's §), actually completing the previous response.
> > > It is worth recalling that a substantial fraction of prompts (22.3% on average) are dropped due to extreme class imbalance prior to the analysis (i.e., when a prompt presents <5 genuine or hallucinated responses).
> > >
> > > |Method|Space|Acc|F1|
> > > |-|-|-|-|
> > > |EigenScore|SBERT 384D|71.0 (27.8)|74.9 (25.9)|
> > > |SelfCheckGPT-NLI|-|82.0 (5.7)|89.8 (3.8)|
> > > |Semantic Entropy|NLI clusters|78.5 (5.5)|86.3 (4.3)|
> > >
> > > Overall, the presented statistics are comparable with the previous results on the reduced dataset.
> > > All the methods perform above random guessing, with SelfCheckGPT presenting the strongest discriminative power and INSIDE's EigenScore the lowest.
> > > In particular, accuracy scores remarkably lower values when compared to F1: this is not surprising, particularly for majority-class baselines considering the overall strong imbalance of the dataset (hallucination rates of 71.9%–90.8% across models).
> > >
> > > ## Q3 follow-up. Ablation
> > > To further strengthen the ablation results we presented in our previous response, we executed both the structural and LP ablation analysis on the CoQA "bridge" dataset (see Rev 4mi3, Q2).
> > >
> > > ### Structural
> > > We report the full ablation results for the structural analysis on our dataset (missing in the original response due to character limit; results slightly differ since we increased the number of randomisations in p-values) and on CoQA.
> > > The percentage of statistically significant (model, prompt) pairs (p<.05) for varying N is reported w.r.t. both the Wasserstein vs. null experiment (W vs W0) and the Wilcoxon test (cf. Fig 2).
> > > The percentage for which W>W0 and the fraction of prompts where the analysis was feasible are reported as well.
> > >
> > > **Ours**
> > >
> > > |N|W vs W0 (p sig) [%]|Wilcoxon (p sig) [%]|Considered samples [%]|W>W0 [%]|
> > > |-|-|-|-|-|
> > > |10| 7.3| 6.1|6.1|79.1|
> > > |15|15.8|18.9|18.9|81.8|
> > > |20|21.3|29.4|29.4|83.2|
> > > |30|30.8|42.7|42.7|84.0|
> > > |50|44.4|56.8|56.8|86.1|
> > > |75|52.8|68.7|68.7|86.6|
> > > |100|58.3|74.5|74.5|87.3|
> > > |150|65.4|83.7|83.7|88.1|
> > >
> > > **CoQA**
> > >
> > > |N|W vs W0 (p sig) [%]|Wilcoxon (p sig) [%]|Considered samples [%]|W>W0 [%]|
> > > |-|-|-|-|-|
> > > |10|30.8|20.4|20.4|87.1|
> > > |15|41.0|36.9|36.9|88.9|
> > > |20|48.6|47.5|47.5|91.4|
> > > |30|58.4|60.4|60.4|92.5|
> > > |50|67.9|72.4|72.4|93.2|
> > > |75|75.1|81.9|81.9|95.0|
> > > |100|77.1|85.5|85.5|95.5|
> > > |150|83.9|91.4|91.4|95.2|
> > >
> > > Surprisingly, the results show even greater separability on the CoQA bridge dataset w.r.t. ours, with above 87% of cases reporting W>W0 even at N=10 (cf. 88.1% at N=150 on our dataset).
> > > The reported significance for both statistical tests is well above 50% within N=30, while at N=150 the significance (particularly for the Wilcoxon test) is remarkable.
> > >
> > > ### Label Propagator (LP)
> > > The impressive separability reported on CoQA from the above ablation is even more appreciable on the LP analysis, with an average accuracy above 90% even when N=10 (F1 = 88.6%).
> > > The full results on both our dataset for reference (cf. Fig 4) and on CoQA are reported in tabular form:
> > >
> > > **Ours**
> > >
> > > |N|F1|Acc|
> > > |-|-|-|
> > > |5|62.5 (19.0)|66.6 (14.0)|
> > > |10|77.9 (14.9)|74.6 (10.9)|
> > > |15|83.1 (12.4)|78.6 (9.9)|
> > > |20|85.7 (11.2)|81.1 (9.5)|
> > > |25|87.0 (10.8)|82.6 (9.3)|
> > > |30|87.9 (10.6)|83.7 (9.1)|
> > > |50|89.7 (9.5)|85.9 (8.7)|
> > > |75|90.3 (9.2)|86.8 (8.4)|
> > > |100|90.5 (8.7)|87.0 (8.1)|
> > >
> > > **CoQA**
> > >
> > > |N|F1|Acc|
> > > |-|-|-|
> > > |5|78.7 (20.2)|82.7 (14.3)|
> > > |10|88.6 (14.2)|91.0 (8.8)|
> > > |15|91.5 (11.9)|93.4 (6.8)|
> > > |20|92.8 (11.4)|94.7 (5.7)|
> > > |25|93.4 (11.3)|95.5 (5.0)|
> > > |30|94.0 (11.1)|96.1 (4.5)|
> > > |50|95.7 (9.2)|97.3 (3.5)|
> > > |75|96.4 (8.5)|97.8 (3.0)|
> > > |100|97.1 (6.4)|98.1 (2.7)|
> > >
> > > Results in terms of both F1 and accuracy (mean and dispersion) are more robust when compared to our dataset.
> > > The increased performance is likely attributable to both the nature of CoQA prompts (shorter, more direct answers that cluster more tightly) and INSIDE's ROUGE-L ≥ 0.5 labelling criterion (cf. Rev 4mi3, Q2), which is more permissive than our LLM-as-a-Judge and potentially regularises the genuine cluster within the embedding space.
> > > These results confirm that LP achieves usable accuracy (>90%) from as few as 10 labelled samples on an independent dataset, validating the practical applicability beyond our original data.
> > >
> > > ---
> > >
> > > We hope this clarifies the remaining points and we thank the Reviewer for the constructive engagement.

---

### Official Review · Reviewer_neJx · 2026-03-12

**Soundness:** 3
**Presentation:** 3
**Significance:** 3
**Originality:** 3
**Overall Recommendation:** 4
**Confidence:** 4

**Summary:**

This paper studies hallucinations in small language models from a geometric perspective. Instead of viewing hallucinations mainly as missing knowledge, the authors hypothesize that when a model generates multiple responses to the same prompt, correct answers tend to cluster tightly in embedding space while hallucinated ones are more dispersed. To test this, they generate 150 responses per prompt across 200 prompts using 10 small LLMs, embed the responses, and analyze the distance distributions between genuine and hallucinated outputs. The analysis shows that correct responses exhibit stronger semantic cohesion and that the two types of responses have distinct geometric patterns in embedding space. Using Fisher Discriminant Analysis, the authors amplify this structural difference to obtain a separable direction between the two classes. They then exploit this geometry to design a label-propagation method that can classify large sets of responses using only a small number of annotated examples (about 30–50), achieving over 90% F1 score in hallucination detection.

**Compliance With Llm Reviewing Policy:**

Affirmed.

**Key Questions For Authors:**

1. The labeling of responses relies primarily on an LLM-as-a-judge (Claude Sonnet) with a relatively small human validation subset (~0.4% of the dataset). How sensitive are your geometric findings and label propagation results to the choice of judge model? For example, do the same geometric separability patterns hold if labels are generated by a different LLM judge or by fully human annotations?

2. Your method analyzes repeated responses to the same prompt and performs label propagation within that prompt’s response set. How well does the geometric signature generalize across prompts or domains? For instance, do Fisher directions or separability patterns transfer across prompts, tasks, or datasets?

3. The proposed framework requires many sampled responses per prompt (e.g., up to 150). In real-world deployment scenarios, models typically produce a single response. Do the authors have insights or preliminary results on whether the geometric signals identified here could be leveraged for hallucination detection in single-response settings?

4. The analysis relies on a specific sentence embedding model (SBERT). How sensitive are the geometric findings and classification results to the embedding space used? Have the authors tested alternative embeddings (e.g., other sentence encoders or model-specific embeddings)?

**Limitations:**

yes

**Strengths And Weaknesses:**

**Strengths**

- The paper provides a clear empirical methodology and large-scale experimental evaluation. The dataset construction (300k responses across 10 small LLMs) and repeated sampling setup are appropriate for studying hallucination variability. The geometric analysis (distance distributions, Wasserstein comparisons, and Fisher Discriminant Analysis) is well motivated and supported by quantitative experiments showing consistent clustering patterns and strong classification performance.

- The paper is generally well structured, with a clear pipeline (dataset generation → structural analysis → label propagation) and figures that effectively illustrate the geometric intuition behind the method (e.g., clustering of genuine vs. hallucinated responses). The methodology and experimental setup are described in a systematic manner that allows readers to understand the proposed analysis framework.

- Hallucination detection and reliability of language models are highly relevant problems. By reframing hallucination as a geometric phenomenon in embedding space, the paper introduces a potentially useful diagnostic perspective that could inspire alternative evaluation tools or lightweight hallucination detection techniques, particularly for small or open-weight models.

- The paper offers a novel conceptual viewpoint: hallucinations are interpreted as retrieval instability reflected in embedding geometry, rather than purely a lack of knowledge. The combination of response-distribution analysis with geometric separability and label propagation from a small annotation set provides an interesting and creative integration of existing tools.

**Weaknesses**

- The evaluation relies heavily on LLM-as-a-judge annotations (Claude Sonnet) with only limited human validation (~1000 samples), which may introduce bias or error in ground-truth labels. Additionally, the analysis focuses on prompts where correct answers are known to exist, which may limit conclusions about hallucinations in broader settings.

- While the narrative is generally clear, the paper occasionally becomes dense with geometric terminology and statistical measures (e.g., Wasserstein distance distributions, Fisher projections) without always providing strong intuition or simplified explanations. Some sections could benefit from clearer interpretation of results and stronger connection between figures and practical implications.

- The practical utility of the proposed detection approach may be limited by its prompt-local assumption (requiring many sampled responses per prompt). In many real-world deployments, only a single response is generated, which makes the approach less directly applicable without modification.

- Although the geometric perspective is interesting, many core components (embedding-based analysis, clustering intuition, Fisher discriminant projections, label propagation) are well-known techniques. The novelty lies primarily in their application and framing, rather than in fundamentally new algorithms or theoretical insights.

---

> ### Author Rebuttal · Authors · 2026-03-31
>
> We thank the Reviewer for the insightful review.
> We are glad the Reviewer appreciated the conceptual viewpoint and the integration of existing tools into a novel analytical framework.
> We address each question below.
>
> ## Q1. Sensitivity to the LLM-as-a-Judge
> The Reviewer raised a very important point.
> Clearly, the LLM-as-a-Judge plays a fundamental role in our study.
> To analyse its impact, we labelled the complete response set for a randomly selected prompt under `gemini-2.5-flash` (1,495 responses), achieving a correlation of 82% with Claude's annotation.
> The Wasserstein ratio, originally equal to 1.15 for both tags, shifted to a ratio in the Fisher space of 4.81 on Claude tag and to 4.70 on Gemini tag.
> This suggests that while individual label flips do occur, the geometric structure is robust to moderate annotation noise.
> We plan to explore multi-judge majority voting in future work. Though, it is worth noting that the 87.9% agreement between Claude and human consensus on 1,000 validated instances (Section D Appendix) provides a solid foundation for the current analysis (Cohen’s κ=.86).
>
> ## Q2. Generalisation of Fisher directions
> We carried out the analysis suggested by the Reviewer.
> We computed Fisher directions independently for each (model, prompt) pair and measured their pairwise cosine similarity.
> Across prompts within the same model, the average cosine similarity is 0.058 (std 0.045), indicating that Fisher directions are largely prompt-specific.
> Across models within the same prompt, the average cosine similarity is 0.145 (std 0.089), showing low transferability.
> Across both models and prompts, the average cosine similarity is 0.055 (std 0.042), showing prompt-specific Fisher directions.
> These results confirm that the discriminative geometry is prompt-conditioned, which is precisely what motivates our local Label Propagation (LP) design rather than a global classifier.
> We view this as a structural insight: the direction separating genuine from hallucinated responses varies with prompt content, reinforcing the view that hallucination geometry reflects the specific knowledge retrieval context rather than a universal model-level property.
>
> ## Q3. Single-response deployment
> We agree that the multi-response requirement limits direct deployment.
> Our current work establishes the geometric foundations: characterising how genuine and hallucinated responses organise in embedding space under repeated sampling.
> We are actively exploring extensions to single-response settings, including graph-based approaches that encode structural invariants from the multi-response analysis into features computable from individual responses.
> We believe the structural characterisation presented here is a necessary prerequisite for such extensions — understanding the geometry of the full response distribution provides the theoretical grounding for designing principled single-response proxies.
>
> ## Q4. Sensitivity to the embedding model
> We definitively agree with the relevance of the sentence encoder choice in our pipeline.
> For this reason, we conducted additional experiments with three alternative encoders (sentence transformers with two different embedding sizes), determining that traceable geometric signatures can be found across all of them.
>
> |Encoder|Dim|W ratio|Fisher sep.|LP F1|
> |-|-|-|-|-|
> |all-MiniLM-L6-v2 (ours)|384|1.13|7.21|86.1%|
> |all-MiniLM-L12-v2|384|1.13|7.38|86.3%|
> |all-mpnet-base-v2|768|1.10|10.47|87.2%|
> |paraphrase-mpnet-base-v2|768|1.11|10.59|87.4%|
>
> Across all encoders, structural separation is preserved, with Wasserstein ratios consistently above 1.10 and 7.2 and LP F1 scores within 1.5% of the original result.
> This confirms that our geometric findings are not an artefact of the specific encoder but reflect genuine structural properties of the response distributions.

---

> > ### Author Rebuttal · Reviewer_neJx · 2026-04-03
> >
> > Thanks for the author response. I have no further questions.

---

> > > ### Author Response · Authors · 2026-04-05
> > >
> > > We sincerely thank the Reviewer for the positive assessment and for confirming that all concerns have been fully resolved.
> > > We are grateful for the constructive engagement throughout this process, which has strengthened the paper.
> > > The additional experiments (encoder sensitivity, Fisher direction generalisation, Gemini cross-validation) will be incorporated into the camera-ready version.
> > >
> > > Given the substantial new evidence provided, which include robustness across four encoders, prompt-specificity of Fisher directions (cosine similarity < 0.06), stability under a different LLM judge (Gemini, 82% agreement, W ratio shift < 2.5%), and cross-dataset replication on CoQA with >97% F1, and provided the acknowledgemnte score of
> > > >Fully resolved - My concerns have been adequately addressed. If you select this option, please consider adjusting your score accordingly.
> > >
> > > we would be grateful if the Reviewer would consider whether an upward revision of the score might be warranted.
> > > We of course fully respect the Reviewer's final judgement.

---

### Official Review · Reviewer_2x73 · 2026-03-13

**Soundness:** 3
**Presentation:** 3
**Significance:** 3
**Originality:** 2
**Overall Recommendation:** 4
**Confidence:** 4

**Summary:**

The paper provides new insights into Hallucination in small scale LLM by comparing the geometry and statistics of the embedding space (using sentence transformer embedding)of Genuine and Hallucinated generated LLM answers of the same prompt. The authors showed that it is possible to propagate a small scale manually annotated sample to large scale of prompt-efficient generation, achieving F1-score above 90%. In addition useful comparison between Hallucinated and Genuine behaviors has been put forward. The authors showed that genuine responses exhibit higher semantic cohesion compared to hallucinated responses, suggesting that embedding-space geometry
can act as a scalable hallucination detection tool. Finally a fully labelled dataset of repeated responses (150) from 200 prompts across 10 small LLMs is released.

**Compliance With Llm Reviewing Policy:**

Affirmed.

**Key Questions For Authors:**

See weakness above

**Limitations:**

yes

**Strengths And Weaknesses:**

Strengths
- The paper is in overall well written with a good literature review
- The method is well explained and motivated
- The results have been validated using a sample of manually annotated dataset
- A new dataset has been released


Weaknesses
- The methodology mentioned the use of three categories Hallucination (H), Genuine (G) and Unknown (U) in case of uncertainty, but the actual study only considers two categories H and G, which put some doubt why the three category-based approach is introduced.
- It is not clear how the 1000 instances were selected for the manual annotation task, whether the classes H and G are equally represented in the this dataset or was it very unbalanced..
- When matching the Claude's label with manual label, it would be interesting to discuss the cases where disagreement occurs to generate some insights for subsequent reasoning.
- The authors only used HugginFace' s sentence encoder as a basis for embedding space and the subsequent statistical and geometrical analysis. This choice has not been well motivated and no comparison was carried out to justify this choice. Especially, as the result leads to highly overlapping distribution, which makes the question whether other embedding space legitimate.
- The authors employed only one single dataset from which they extracted events from 2020 to 2022. This adds extra uncertainty as the way events extracted using topic modelling as in Pezik et al.'s paper can add extra uncertainty. It would be more interesting if some publicly available dataset were used to ease comparison and identify useful baseline models.
- The authors have not provided any alternative results or methods to exhibit the efficiency of the developed model with respect to state of the art.
- The authors reported results with stemmed responses; however, it is unclear whether the parser stemming can preserve semantic content as claimed by the authors, as it usually looses semantic content. No evaluation was carried out to test this hypothesis.

---

> ### Author Rebuttal · Authors · 2026-03-31
>
> We thank the Reviewer for the detailed and constructive feedback.
> We address each point below.
>
> ## W1. Unknown (U) category.
> The U label serves as a fallback for responses where the LLM-as-a-Judge cannot determine factual correctness with confidence — typically hedged answers, refusals to answer, or responses that mix correct and incorrect content.
> A total of 2,205 responses (0.735% of the full dataset) were labelled U and excluded from all subsequent analyses.
> We manually inspected 441 (20%) of the U responses and confirmed they predominantly correspond to: (a) explicit knowledge-cutoff refusals, where the model cites a specific training cutoff date as the reason for not answering; or (b) generic epistemic refusals, where the model acknowledges ignorance without a temporal justification and typically redirects the user to external sources.
> We will further clarify this role in the camera-ready.
>
> ## W2. Selection of the 1000 annotated instances.
> The 1000 instances were selected via stratified random sampling, ensuring representation across models and prompts.
> The class distribution in the annotated set is 15.7% Genuine and 84.3% Hallucinated, which is in line with the distributions provided in Table 4 of the Appendix.
> We will report this explicitly in the revised manuscript.
>
> ## W3. Analysis of Claude–human disagreements.
> We definitively agree with the Reviewer that this analysis is informative.
> Of the 1000 annotated instances, 12.1% showed disagreement between Claude and the human consensus (11.7% false hallucinations, 0.4% false genuines).
> We categorised these into 5 main types: (i) accurate responses with minor imprecisions where Claude was stricter, (ii) ambiguous temporal boundaries, (iii) correct responses penalised for useless prompt repetition, (iv) correct responses with additional unrequested information, and (v) lack of access to the correct answer.
> We will include a dedicated appendix discussion in the camera-ready.
>
> ## W4. Embedding encoder sensitivity.
> We fully agree that the contribution would benefit from a more detailed discussion on this topic.
> For this reason, we compared the results with three alternative sentence encoders, finding that the geometric structure and Label Propagation (LP) performance remain consistent across all of them.
> We provide full details and a comparative table in our response to Reviewer neJx (Q4), to which we kindly refer the Reviewer.
>
> ## W5. Dataset choice.
> To the best of our knowledge, no publicly available dataset provides large amounts of repeated tagged responses per prompt across multiple models, which is a prerequisite for our distributional analysis.
> Existing hallucination benchmarks typically provide single responses per prompt.
> Our dataset release is intended precisely to fill this gap.
> Additionally, we have replicated our structural analysis on a dataset we have generated upon responses provided by CoQA dataset (a benchmark also used by Chen et al., ICLR 2024), confirming that our geometric findings generalise beyond our dataset; we discuss this in detail in our response to Reviewer 4mi3.
>
> ## W6. Comparison with state of the art.
> We would like to clarify that our primary contribution is the structural geometric characterisation of hallucination, with LP serving as a practical downstream application.
> We have now included comparisons with standard classifiers (logistic regression, SVM, kNN, nearest centroid) as well as the EigenScore metric (Chen et al., ICLR 2024), Semantic Entropy, and SelfCheckGPT on the same data and evaluation protocol (see details in our response to Reviewer 82cu, Q2).
>
> ## W7. Stemming.
> The main results in the paper use complete (unstemmed) responses, and the stemmed results in the Appendix show consistent performance (accuracy differences ≤0.3% across all models), confirming that the classification framework operates on semantic structure rather than lexical artefacts.
> To directly assess semantic preservation, we computed the cosine similarity between plain and stemmed response embeddings, obtaining a mean of 93% (std 5%) across all responses, actually confirming that stemming retains the vast majority of the information our framework operates on.
> We will reframe the stemming analysis as a robustness check and defer a full evaluation of stemming's impact on semantic preservation to future work.

---

> > ### Author Rebuttal · Reviewer_2x73 · 2026-04-03
> >
> > I acknowledge the response from the authors.
> > The response W3 is not fully addressed, claiming the responses will be in the new appendix of the revised version is not serious, we would expect at least to see some result table here.
> > Apart from that the responses are technical sound. The score is already on positive note, so maintained

---

> > > ### Author Response · Authors · 2026-04-05
> > >
> > > We thank the Reviewer for acknowledging that the remaining six points are technically sound.
> > > We sincerely apologise if the W3 response came across as dismissive: this was definitely not our intention.
> > > Both the temporal and character limits forced us to prioritise the new experimental results (baseline comparisons, encoder sensitivity, cross-dataset replication on CoQA, structural ablation), which collectively consumed the vast majority of the available space and time.
> > > We fully understand the Reviewer's expectation and are happy to provide the complete analysis here.
> > >
> > > ## Detailed analysis of Claude–human disagreements
> > >
> > > Of the 1,000 manually annotated instances, 121 (12.1%) showed disagreement with Claude's labels.
> > > The asymmetry in the disagreement is relevant: 117 false hallucinations (Claude labelled H, human consensus G) and only 4 false genuines (Claude labelled G, human consensus H).
> > > This indicates that Claude errs almost exclusively on the conservative side (i.e. flagging correct responses as hallucinated) rather than missing actual hallucinations.
> > >
> > > We categorised all the 121 disagreement cases into five types, summarised below:
> > >
> > > |Type|Count|% of disagreements|Description|
> > > |-|-|-|-|
> > > |(i) Minor imprecisions|27|22.30%|Response is substantively correct but contains a minor inaccuracy (e.g., approximate date, slightly imprecise name, wrong minor detail); Claude penalises strictly while human annotators judge the core answer as correct|
> > > |(ii) Temporal ambiguity|12|9.96%|The question targets a specific year, but the correct answer spans an ambiguous boundary (e.g., an event announced in late 2019 and completed in 2020), or simply the model answered with a correct fact which did not happen in the requested period|
> > > |(iii) Prompt repetition|11|9.08%|The response contains the correct answer but also repeats or paraphrases the original question; Claude treats the repetition as a sign of low-quality generation and flags it as hallucinated|
> > > |(iv) Unrequested information|40|33.05%|Response provides the correct answer alongside additional (unrequested but factually correct) information; Claude flags the extra content as potentially fabricated|
> > > |(v) Inaccessible ground truth|31|25.61%|Claude consistently labels correct answers to the same prompt as incorrect (or vice-versa), likely due to limited access to the ground-truth information|
> > >
> > > **Key observation:** types (i), (iii), and (iv) all reflect Claude being *stricter* than human annotators (~65%), i.e. offering a conservative bias that makes our hallucination labels slightly pessimistic (underestimating the genuine rate).
> > > This bias is favourable for our structural analysis: it means any geometric separation we observe between G and H offers a lower bound on the true separation, since some responses labelled H are actually correct.
> > >
> > > ## Examples of misclassifications
> > > We report one sample (trimmed due to space) per each of the five classess that where misclass as non-hallucinated:
> > > ```
> > > Q: Which golfer won the PGA Championship in 2022?
> > >
> > > A: The answer is Justin Thomas. He shot an eight-under par total of six under at Southern Hills Country Club, Oklahoma on Sunday morning (AEDT). The win was his second major title after he had previously claimed victory back when it took place last year’s US Open tournament [...]
> > >
> > > TYPE: (i) Minor imprecisions (wrong par count and first prize)
> > > ```
> > >
> > > ```
> > > Q: Who won the Daytona 500 in 2022?
> > >
> > > A: The winner of the Daytona 500 race held on February 19th, 2023 was Austin Cindric driving for Team Penske [...]
> > >
> > > TYPE: (ii) Temporal ambiguity
> > > ```
> > >
> > > ```
> > > Q: Which artist won Grammy Song of the Year in 2020?
> > >
> > > A: - The answer is “Bad Guy” by Billie Eilish. She was also nominated for Record of the year, Best Pop Solo Performance, Album of the Year, and Best New Artist.\n\nExplain your response with as much detail as possible. Use specific examples from the text or videos we’ve watched this week to support your ideas. You can use [...]
> > >
> > > TYPE: (iii) Prompt repetition
> > > ```
> > >
> > > ```
> > > Q: Which film won the Golden Bear at Berlin in 2022?
> > >
> > > A: "The answer is very simple, it's "Alcarràs"".
> > > But what if you don't know this title yet but have heard of another one that has been awarded an award or prize during these past months. I am talking about [...]
> > >
> > > TYPE: (iv) Unrequested information
> > > ```
> > >
> > > ```
> > > Q: Which film won the BAFTA Award for Best Film in 2020?
> > >
> > > A: "The winner of the BAFTA Award for Best Film in 2020 was ""Parasite."""
> > >
> > > TYPE: (v) Inaccessible ground truth (In all the analysed response Parasite was flagged as correct despite 1917 is the correct answer. Parasite won "Best Film Not in the English Language")
> > > ```
> > >
> > > ---
> > >
> > > We will include the full table, the discussion and the samples as a dedicated appendix in the camera-ready version.
> > > To further clarify, all of the other responses will be included as part of the final manuscript as well.
> > > We hope this addresses the Reviewer's concern in full and shows that the analysis was already completed without lacking in seriousness.

---

### Decision · Program_Chairs · 2026-04-30

**Decision:**

Accept (regular)

**Comment:**

This paper studies hallucination in small language models through a geometric lens, arguing that repeated genuine responses to the same prompt exhibit tighter cohesion in embedding space than hallucinated ones, and leverages this structure for label propagation from a small annotated set. Reviewers agreed that the problem is interesting and that the paper provides a substantial empirical study, including a large repeated-sampling dataset across 10 models, clear structural analyses, and a useful released resource. The main concerns were about the strength of the novelty relative to prior work such as INSIDE, reliance on LLM-as-a-judge labels, the practical limitation of requiring many responses per prompt, and initially missing baseline and external-dataset comparisons. The rebuttal substantially strengthened the paper by clarifying that the contribution is primarily structural rather than a formal proof, adding broad baseline comparisons, reporting sample-size ablations, showing robustness across multiple encoders and a second judge, and providing cross-dataset replication on CoQA. While I agree that the novelty is moderate and that the method’s direct deployment utility is limited by its prompt-local repeated-sampling assumption, I find the paper technically sound, empirically strengthened after rebuttal, and likely to be useful to the community as an analytical perspective and dataset resource. On balance, I recommend accept.